# Can discharge be used to inversely correct precipitation?

Ashish Manoj J[1, a], Ralf Loritz[1], Hoshin Gupta[2], Erwin Zehe[1]

1 Chair of Hydrology, Institute of Water and Environment (IWU), Karlsruhe Institute of Technology, 76131 Karlsruhe, Germany

2 Department of Hydrology and Atmospheric Sciences, The University of Arizona, Tucson 85721, United States

[a] *Correspondence to: Ashish Manoj J ([ashish.jaseetha@kit.edu](mailto:ashish.jaseetha@kit.edu))*

**Abstract.** This study explores the feasibility of using the information contained in observed streamflow measurements to inversely correct catchment-average precipitation time series provided by reanalysis products. We explore this possibility by training LSTM models to inversely predict precipitation. The first model uses discharge as an input feature along with other meteorological factors, while the second model uses only the meteorological factors. Although the model provided with discharge information showed better mean performance, a detailed analysis of various time series measures across the continental scale revealed underestimation biases when compared with the original reanalysis product used for training. However, an out-of-sample test showed that the inversely estimated precipitation is better able to reproduce small-scale, high-impact events that are poorly represented in the original reanalysis product. Further, using the inversely generated precipitation time series for classical hydrological "forward" modeling resulted in improved estimates for streamflow and soil moisture. Given the notable disconnect between reanalysis products and extreme events, particularly in data-scarce regions worldwide, our findings have implications for achieving better estimates of precipitation associated with high-impact events.

## 1 Introduction

The performance of hydrological models has traditionally been constrained by the availability and quality of observations covering various aspects of the water cycle. Among those, precipitation and streamflow observations are pivotal, as they represent cause-and-effect in the context of system dynamics. Long-term experimental data from well-studied research catchments, and data from operational monitoring networks, have thus long been the cornerstone of the hydrological sciences (Tetzlaff et al., 2017). The relevance of observed data and research observatories cannot be overemphasised, particularly due to the invalidity of stationarity assumptions (Milly et al., 2008) in the face of anthropogenic climate change and its impacts on water-related hazards and availability.

As the availability and quality of observations crucially constrain the "realism" of a hydrological model and thus the accuracy of predictions, data scarcity impedes accurate modelling and inference of hydrological processes. Global reanalysis products (Muñoz-Sabater et al., 2021; ONOGI et al., 2007; Rienecker et al., 2011) can potentially, if of sufficient quality, complement the few existing ground-based observations by offering a valuable alternative when exhaustive local observations are not available. Further, they play a pivotal role in hydro-climatic research (Alexopoulos et al., 2023; Gu et al., 2023), by providing a consistent, long-term view of the state of the global climate system via the assimilation of measurements and monitoring data into numerical weather models.

While previous studies (Essou et al., 2016; Tarek et al., 2020) have already shown the value of using reanalysis data as estimates for meteorological forcing data in regions with little or sparse ground-based weather station data, serious concerns about their quality remain when used in the context of hydrological modelling. The main issues include (Tarek et al., 2020) (i) regional variations in data quality and (ii) limited representation of local hydro-meteorological processes, with both of these impacting/biasing model structures and simulated states and fluxes. Systematic biases are also critical obstacles to the broader

applicability of such products (Clerc-Schwarzenbach et al., 2024). In the case of ERA5-Land, a component of the Copernicus Climate Change Service (C3S) provided by the European Centre for Medium Weather Forecasting (Muñoz-Sabater et al., 2021), there is a tendency to significantly overestimate potential evapotranspiration (Clerc-Schwarzenbach et al., 2024; Kratzert et al., 2023; Xu et al., 2024). Deficiencies have also been documented in the representation of convective storms (Essou et al., 2016; Taszarek et al., 2021) with subsequent underestimation of precipitation magnitudes and intensities (Manoj

J et al., 2024).

It is important to stress that "true" precipitation estimates are per default unknown at the catchment scale. We obtain estimates of them (with considerable uncertainty) by either interpolating data from stations in or surrounding the catchment or averaging gridded data from reanalysis/remote sensing products to the catchment scale. Such precipitation uncertainty is rarely considered when quantifying model output uncertainty; while studies are usually conducted to show how differences in

simulated discharge can be as a consequence of changing precipitation input, they rarely look at how much improvement of the model performance would be possible by using different but plausible precipitation (Bárdossy et al., 2022, 2020).

Because precipitation forcing data plays a crucial role in rainfall-runoff modelling, several methods (Yumnam et al., 2022) have been suggested for correcting precipitation data. These range from the use of storm multipliers (Sun and Bertrand-Krajewski, 2013) to station-wise correction of data using a gauge-based precipitation network (Cornes et al., 2018). However,

gauge-based methods require a sufficient number of weather stations (Agarwal et al., 2020), which is often not the case for most regions around the world. As seen from previous experience, the observation network is too sparse even in data rich regions, and the majority of high-impact rainstorms are simply not observed (Borga et al., 2008). This is particularly true for flash floods in response to convective storm activity (Manoj J et al., 2024; Meyer et al., 2022; Villinger et al., 2022) and well related to the classical "Predictions in Ungauged Basins - PUB problem" (Sivapalan et al., 2003). To overcome this problem,

and in line with Kirchner's (2009) work on "doing hydrology backwards", this paper explores options for inverse estimation of precipitation using the information contained in observed streamflow. The goal is to determine whether inverse estimation at the catchment scale can refine precipitation estimates from reanalysis products, ensuring they are hydrologically consistent, especially for extreme events.

While the classical "forward rainfall-runoff generation problem" has received considerable attention over various decades

(Montanari et al., 2013; Sivapalan et al., 2003), a smaller subset of studies (Brocca et al., 2013; Kirchner, 2009; Kretzschmar et al., 2014; Krier et al., 2012; Teuling et al., 2010) has investigated the feasibility of tackling the inverse problem efficiently. Kirchner (2009) reported an early and successful attempt to infer catchment average rainfall and evaporation time series from streamflow fluctuations and inspired several investigations examining the advantages and limitations of doing 'hydrology backwards' in diverse catchments (Krier et al., 2012; Teuling et al., 2010). Although these studies have established a robust

mathematical foundation for addressing the inverse hydrological problem, they were limited to smaller, well-monitored research catchments. This raises questions about the applicability of this approach to larger catchments as well as to smaller, non-experimental ones.

Note that inversions of the catchment water balance are inherently ill-posed, making it near impossible to find a unique solution (Bishop, 2006). Adopting the concept of micro- and macro-states from statistical mechanics (Zehe and Blöschl, 2004), we

argue that the exact micro-state, i.e. the "true" space-time pattern of precipitation in the catchment, is neither uniquely identifiable nor observable. Yet, we conjecture that streamflow data can reduce the uncertainty associated with this process, because it provides valuable information on antecedent precipitation. As streamflow remains a non-linear convolution of the catchment-average precipitation, we propose that machine learning is well suited to this problem. Deep learning has recently revolutionised almost all fields of the natural sciences and engineering, showing great promise in solving a wide range of

inverse problems, especially those related to imaging (Ongie et al., 2020). It has also been argued that such models can provide

meaningful and general benchmarks for hypothesis testing (Klotz et al., 2022; Nearing and Gupta, 2015) and afford powerful avenues for generalisation using large datasets (Loritz et al., 2024b).

The overall objective of this study is to 'do hydrology backwards' using a regional-scale long short-term memory (LSTM) network model trained on large-scale hydrological datasets using the ERA5 Land precipitation product (Muñoz-Sabater et al., 2021) as a target. While ERA5 Land has well-documented issues in representing the driving precipitation estimates for specific event scales (Essou et al., 2016; Manoj J et al., 2024), recent studies (Bandhauer et al., 2022; Goteti and Famiglietti, 2024) have shown that they hold considerable promise to tackle the "Predictions in Ungauged Basins - PUB problem". The underlying research question is, "How much information about the catchment-average precipitation is effectively encoded in the variability of the streamflow time series observed at the outlet?" To answer this question, we first investigate whether the approach can accurately replicate the spatial characteristics of the original forcing reanalysis dataset (by looking at various time series measures) across European catchments for an unseen testing period. We then examine how the inverse model performs when moving to much smaller (50-200 km$^2$: Table 2) out-of-sample catchments. Here, we compare (using the event runoff coefficients) LSTM-based inverse estimates during flood events to the original reanalysis product (ERA5 Land) and rain gauge-based observational estimates over the same region (E-OBS: Cornes et al., 2018). Finally, we use the HBV conceptual hydrological model (Bergström and Forsman, 1973) and the spatially-distributed, process-based CATFLOW model (Zehe et al., 2001) to assess the quality of the precipitation estimates for forward modelling of streamflow and soil moisture dynamics, respectively.

## 2 Data and Methods

### 2.1 Model Configuration

LSTMs (Hochreiter, 1998) are a special type of recurrent neural network that makes use of cell states and so-called 'gates' to control the information flow through the network. The LSTM model used in this study extends upon the work of Kratzert et al. (2018) and Acuña Espinoza et al. (2024). The LSTM architecture, which is commonly used for streamflow simulation in hydrology (Kratzert et al., 2018) uses a sequence of meteorological variables, such as precipitation and temperature as dynamic inputs, along with catchment attributes as static features, to predict the corresponding streamflow. In our setting, to establish an inverse model, we use the same general model architecture as in previous studies (Acuña Espinoza et al., 2024; Loritz et al., 2024b). The key difference is that future streamflow is now used along with other dynamic and static data as inputs (Table A1 in Appendix A) in order to estimate the precipitation forcings of the catchments. To account for the time lag between precipitation and discharge response observed at the catchment outlet, the model was provided with a 7-day lead time series for discharge. We explored ranges of hyperparameter settings on a smaller subset of the training dataset to establish relatively stable hyperparameter configurations (Fig. S1 in Supplementary Information), finally setting them according to (Acuña Espinoza et al., 2024) with a reduced number (5) of training epochs. Table A2 in Appendix A indicates the values used for the LSTM network hyperparameters. Mean squared error was used as the training loss function. The codes for model building and training can be found online (Manoj J, 2025a) . The LSTM was trained as a regional model (single network trained on all available catchments) based on the openly available datasets detailed in the next section (Section 2.2). For forward hydrological modelling using the inversely-generated precipitation timeseries estimates, we use two hydrological models (Appendix B) - the lumped conceptual HBV model (Hydrologiska Byråns Vattenbalansavdelning: Bergström and Forsman, 1973) and the spatially distributed process-based CATFLOW model (Zehe et al., 2001).

### 2.2 Data sets

This study utilized the Caravan dataset (Kratzert et al., 2023) to investigate our hypothesis regarding the inverse identifiability of precipitation from information about discharge dynamics. We trained our model on European catchments from the GRDC-Caravan (Färber et al., 2023) community extension and the original Caravan dataset, which includes

catchments from CAMELS-GB (Coxon et al., 2020). The Caravan dataset uses the ERA5 Land (Muñoz-Sabater et al., 2021) as meteorological forcing, while the catchment attributes include data from HydroATLAS (Linke et al., 2019). The discharge data is tapped from relevant state and national authorities and is accessible as open datasets. Figure S2 in the Supplementary information depicts the study catchments (1804 in total) in the training dataset.

We chose a training period of around 25 years between 01 October 1980 to 30 September 2005. Following the best practices in data-based modelling, the model was tested on an unseen testing period between 2006 and 2020 (2015 for CAMELS-GB catchments due to data unavailability). To investigate its generalizability across scales, we also tested the model on four catchments (Fig. S3 & S4) that were not included in the original training set (Section 2.3.2). For the out-of-sample test, we made use of data from the Caravan Spain (Casado Rodríguez, 2023) and Caravan Switzerland (Höge et al., 2023) extensions, in addition to data from local data providers in Germany (Landesanstalt für Umwelt, Messungen und Naturschutz Baden-Württemberg—LUBW) and Luxembourg (Nijzink et al., 2024). The observational E-OBS precipitation product (v27.0 - Cornes et al., 2018), which uses the station network of the European Climate Assessment & Dataset (ECA&D) project, was used as another benchmark for the evaluation of model performance. To validate the inversely generated precipitation (Section 2.3.3) during forward modeling, we conducted hydrological model simulations in the Elsenz Schwarzbach and Lippe catchments (Fig. S5). Table 1 provides an overview of the datasets used in this study, detailing their spatial and temporal resolutions, as well as their sources.

**Table 1 Brief overview of the datasets used in this study, including their spatial and temporal resolution.**

| Dataset | Type & Source | Spatial Resolution | Temporal Resolution | Details |
|---|---|---|---|---|
| Caravan | Hydrometeorological dataset (Kratzert et al., 2023) | Catchment scale | Daily | Open community dataset that includes catchment forcing data and attributes along with streamflow. |
| GRDC-Caravan | Hydrometeorological dataset (Färber et al., 2023) | Catchment scale | Daily | Community extension to the Caravan dataset, incorporating data from the Global Runoff Data Centre (GRDC). |
| ERA5 - LAND | Reanalysis product (Muñoz-Sabater et al., 2021) | 0.1º x 0.1º | Hourly (aggregated to daily) | Reanalysis product produced by replaying the land component of ERA5 climate reanalysis |
| E-OBS | Gridded observational precipitation product (Cornes et al., 2018) | 0.1º x 0.1º | Daily | Interpolated observational precipitation product utilizing the station network from the European Climate Assessment & Dataset (ECA&D) project. |
| Caravan Spain | Hydrometeorological dataset (Casado Rodríguez, 2023) | Catchment scale | Daily | Community extension to the Caravan dataset, incorporating data from Spain. |
| Caravan Switzerland | Hydrometeorological dataset (Höge et al., 2023 | Catchment scale | Daily | Community extension to the Caravan dataset, incorporating data from CAMELS-CH catchments. |
| Caravan Germany | Hydrometeorological dataset (Dolich et al., 2025) | Catchment scale | Daily | Community extension to the Caravan dataset, incorporating data from CAMELS-DE catchments. |
| MERRA-2 | Reanalysis product (Gelaro et al., 2017) | 0.625º x 0.5º | Hourly (aggregated to daily) | Global atmospheric reanalysis by NASA Global Modeling and Assimilation Office (GMAO) using the Goddard Earth Observing System Model (GEOS) |
| GLDAS-2.2 | Reanalysis product (Li et al., 2019) | 0.25º x 0.25º | Daily | NASA Global Land Data Assimilation System model outputs with data assimilation for the Gravity Recovery and Climate Experiment (GRACE-DA) |

 **2.3 Experimental Design**

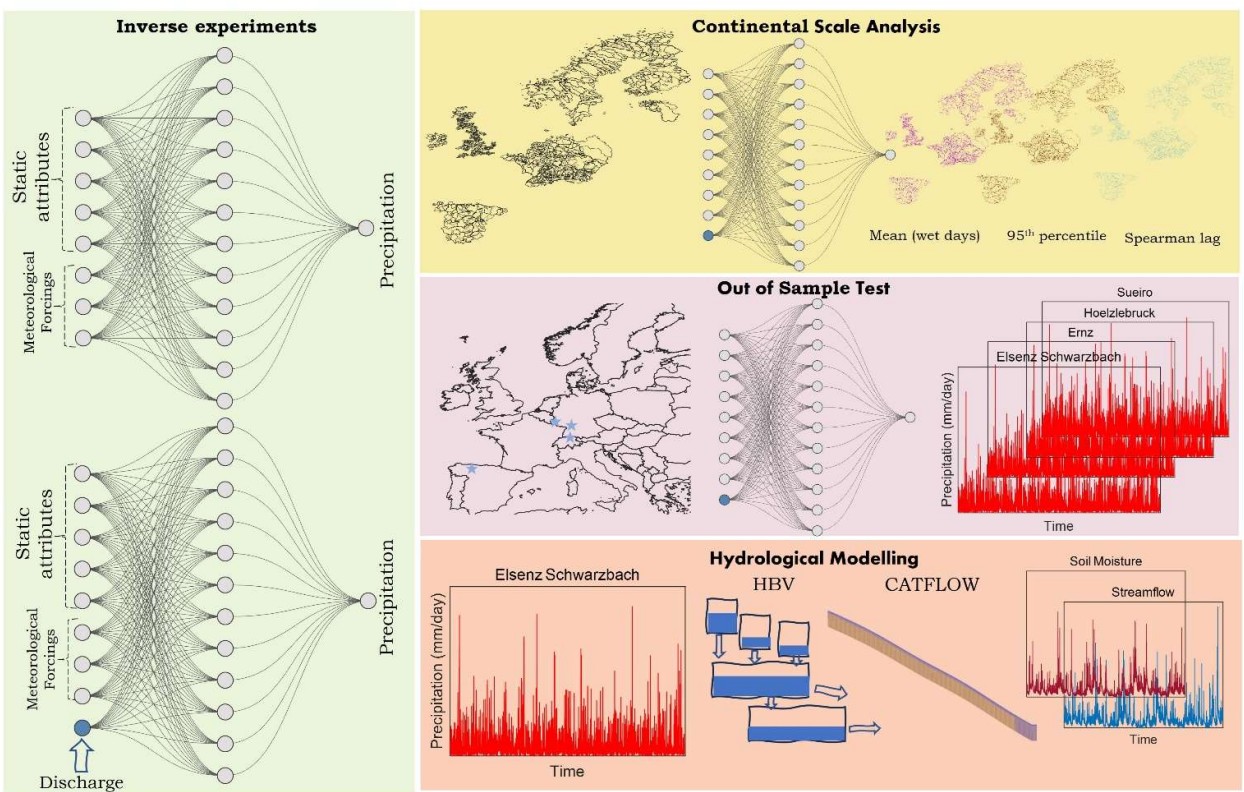

**Figure 1 Schematic representation of our methodological approach. Each rectangular panel indicates different stages of our workflow. Initially, we train two LSTM models to predict catchment average precipitation through inverse experiments (Section 2.3.1). The trained model with discharge data (*with_discharge*) is then utilized for a continental-scale analysis before being used for**
**out-of-sample testing (Section 2.3.2). Finally, a validation exercise for the inversely generated precipitation is conducted using various hydrological models (Section 2.3.3).**

**2.3.1 Exploring information about precipitation in streamflow**

To shed light on the value of discharge for inversely predicting precipitation, we conducted a virtual experiment (Fig. 1) in which two LSTM models (Tables A1 and A2 in Appendix A) were trained using the same catchments and training period.
The first model (*without_discharge*) used only meteorological time series (air temperature, solar and thermal radiation) and static attributes (area, p_mean, ele_mt_sav, frac_snow, pet_mm_syr: Kratzert et al., 2023), while the second model (*with_discharge*) included lagged discharge as an additional input variable. Both models were trained to predict daily catchment average precipitation sums (ERA5 Land). Therefore, we only deal with spatially averaged timeseries for precipitation, assuming that these values represent the effective precipitation over the entire catchment.

We then used the trained regional-scale model (*with_discharge*) to predict the precipitation time series inversely for all the test catchments over the unseen testing period and evaluated (Appendix C) those using the mean wet day precipitation (MWD) – mm/day, 95th percentile limit (R95P) – mm/day, and Spearman autocorrelation values (SL) for each catchment, and then compared them to the values from ERA5 Land (used for training the model) and E-OBS (observational product) at the continental scale.

**2.3.2 Out of sample precipitation inversions and their quality**

We further tested the feasibility of knowledge transfer to out-of-sample catchments and used the same regional-scale model (*with_discharge*) to inversely predict the intensity of driving rainstorms for selected flood events in four hydro-

climatically diverse and much smaller catchments (not included in the original training dataset). These catchments (Table 2 & Fig. S3-S4) were chosen based on the severity of the flooding and on the apparent inability of ERA5 Land forcings to accurately

represent the storms that triggered the flood events.

**Table 2 Attributes for the four catchments used for out-of-sample testing.**

| Catchment | Country | Area (km$^2$) | Mean precipitation (mm/day) | Mean potential evapotranspiration (mm/year) | Mean elevation (m) |
|---|---|---|---|---|---|
| Elsenz-Schwarzbach | Germany | 196.5 | 2.51 | 812.85 | 246.7 |
| Ernz | Luxembourg | 69.3 | 2.31 | 724.04 | 345.5 |
| Sueiro | Spain | 132.5 | 3.31 | 873 | 381 |
| Hoelzlebruck | Germany | 47.1 | 4.14 | 658 | 980 |

### 2.3.3 The potential of inverted precipitation for forward modelling

To evaluate the value of generated precipitation data for forward modeling of streamflow, we calibrated the HBV conceptual
hydrological model (Bergström and Forsman, 1973) over the Elsenz Schwarzbach (Manoj J et al., 2024)  and Lippe (camelsde_DEA11130: Loritz et al., 2024a) catchments (Fig S5 in Supplementary) using both the original ERA5 Land and the *with_discharge* LSTM-generated precipitation timeseries and compared the evaluation period performance of both model versions (Table B1 in Appendix B). The HBV model (Appendix B) used in this paper requires precipitation (ERA5 Land/LSTM simulated), potential evapotranspiration, and air temperature as inputs. We follow the recommendations of Clerc-
Schwarzenbach et al. (2024), similar to that of Loritz et al (2024), for the calculation of potential evapotranspiration, and use the temperature-based Hargreaves formula detailed by Adam et al. (2006).

Complementary to streamflow modelling, the performance of a hydrological model can also be judged by how well it replicates the catchment dynamics of a region. Soil moisture is a key variable controlling the partitioning of net radiation into sensible and latent heat (Seneviratne et al., 2010) or overland flow during a rainstorm (Zehe and Blöschl, 2004). We thus used each
precipitation estimate (*with_discharge* LSTM and ERA5 Land) to run the process-based hillslope scale model CATFLOW (Appendix B), using a setup from Manoj J et al. (2024) used for uncalibrated predictions of local floods. Here, we focused on one of the headwater sub-catchments (Catchment W32 in Fig. S5) within the Elsenz Schwarzbach. The model simulated (Table B1) the period from 01 January 2008 to 31 December 2015 using each of the ERA5 Land and *with_discharge* LSTM precipitation estimates, and the corresponding spatially averaged soil moisture states were compared against several soil
moisture reanalysis products (Table 1: due to the unavailability of observed data). These include a) ERA5 Land: Muñoz-Sabater et al., 2021) b) GLDAS (NASA Global Land Data Assimilation System, GLDAS-2.2 GRACE DA: Li et al., 2019) and c) MERRA (Modern-Era Retrospective analysis for Research and Applications version 2 – tavg1_2d_lnd_Nx: Gelaro et al., 2017)


## 3 Results

### 3.1 The information contained in streamflow about precipitation

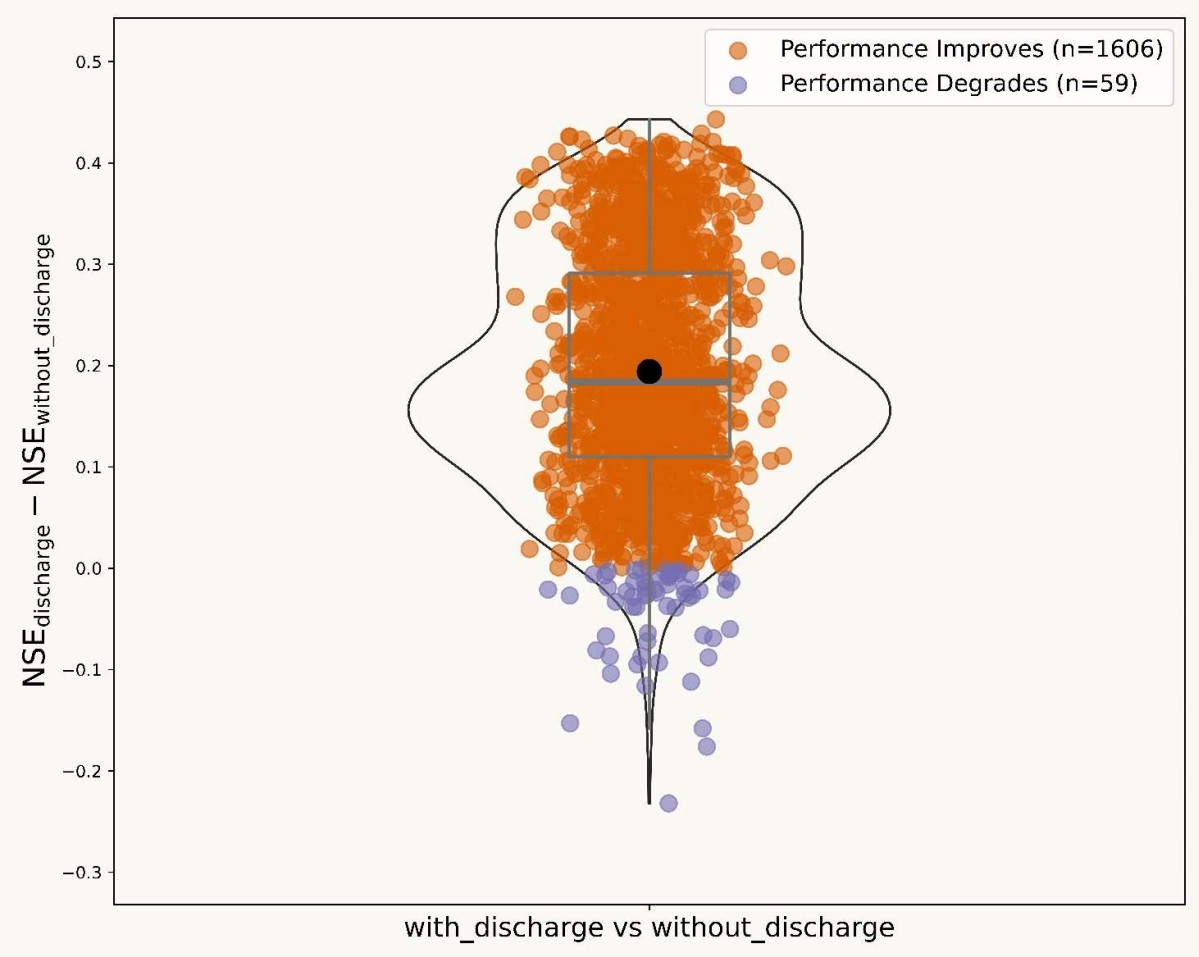

**Figure 2 Violin plot displaying the pairwise differences (*with_discharge* vs *without_discharge* models) in NSE for the study catchments.**

Figure 2 shows a violin plot displaying the pairwise difference in the mean performance of the two LSTM models (Figure A1 in Appendix A) over the catchments in the test dataset. Each point denotes the difference in NSE (Appendix C) for individual catchments while making predictions using the *with_discharge* model compared to the *without_discharge* model. A marked shift towards higher positive differences indicates that the model "*with_discharge*" has higher NSE values than the model "*without_discharge*". This holds true not only on average but also with respect to the best-performing catchments (n=1606 in total). The median NSE metric value ( Nash and Sutcliffe, 1970) for the regional LSTM model across the study catchments is about 20% higher when discharge is used as an additional predictor than when it is not. However, it is also observed that in a few cases (n=59), discharge information has worsened the performance – likely due to the poor quality of streamflow data in these catchments.

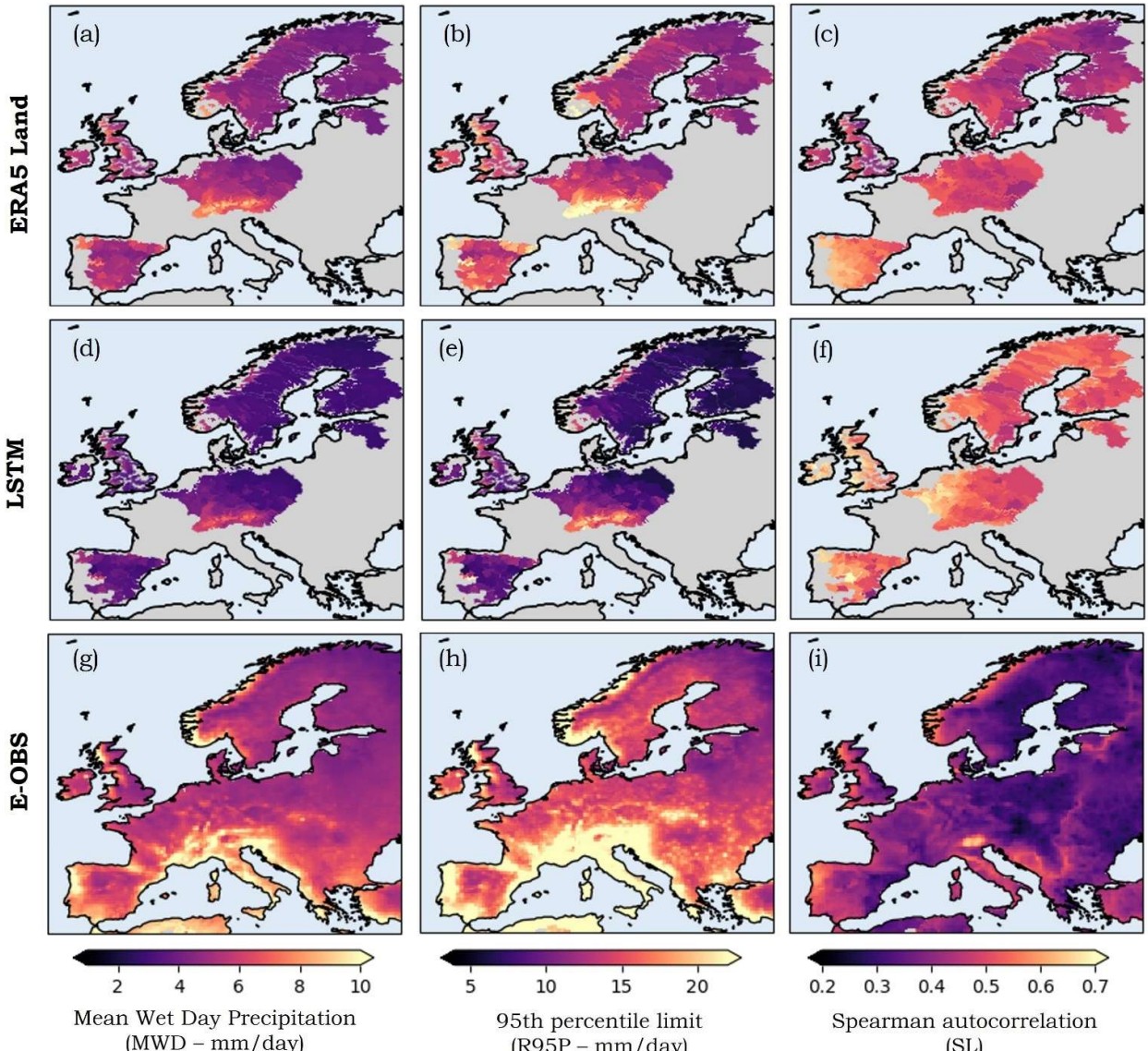

**Figure 3 The spatial patterns of the different time series metrics (Appendix C) mean wet day precipitation (MWD) – mm/day, 95th percentile limit (R95P) – mm/day, and Spearman autocorrelation values (SL)  over the study catchments for the three different sources ERA5 Land (top row): a) to c),** *with_discharge* **LSTM model (middle row): (d) to (f) and E-OBS (bottom row): (g) to (i) from 2006 to 2020 (2015 for CAMELS-GB catchments).**

To examine the characteristics of the simulated time series from the *with_discharge* model over the testing period in detail, we computed three timeseries measures (Appendix C) mean wet day precipitation (MWD) – mm/day, 95th percentile limit (R95P) – mm/day, and Spearman autocorrelation values (SL) across all the catchments, and show the results in Fig. 3.

The continental-scale analysis reveals distinct patterns for the major European climatic regions. The spatial patterns for the mean wet day precipitation (Fig 3a-g: MWD) obtained using the *with_discharge* LSTM model are well aligned to the ones from ERA5 Land and EOBS. Higher daily average values are observed towards the Alps, the Carpathian Mountain ranges, and the coast of Norway, consistent with the climatology of these regions. However, the model systematically underestimates absolute values, as evident from the scatterplot shown in Fig. 4.

A comparison with the total daily means (including both rainy and non-rainy days; Fig. S6 in Supplementary) shows that this underestimation is particularly severe while considering only rainy days (daily precipitation >1 mm). For the 95th percentile of wet days (R95P), we again see a robust representation of the spatial differences, along with an underestimation of the

magnitudes (Fig. 3b-h). The Spearman autocorrelation coefficient values (SL: Fig 3c-i) indicate that while the model underestimates the mean and 95th percentile limits, it overestimates the lag coefficient (which indicates the persistence in the precipitation time series) compared to the ERA5 Land time series. In addition, we also see that the ERA5 Land largely matches with the precipitation field's characteristics (wet day mean and 95th percentile limit) as in the observational E-OBS product.

The higher autocorrelation values for both *with_discharge* and ERA5 Land may arise from model products incorporating catchment persistence, unlike the gridded observational E-OBS data. In the case of the *with_discharge* LSTM model, the even higher values are likely due to the inclusion of highly auto correlated streamflow data, which adds redundancy or a longer memory.

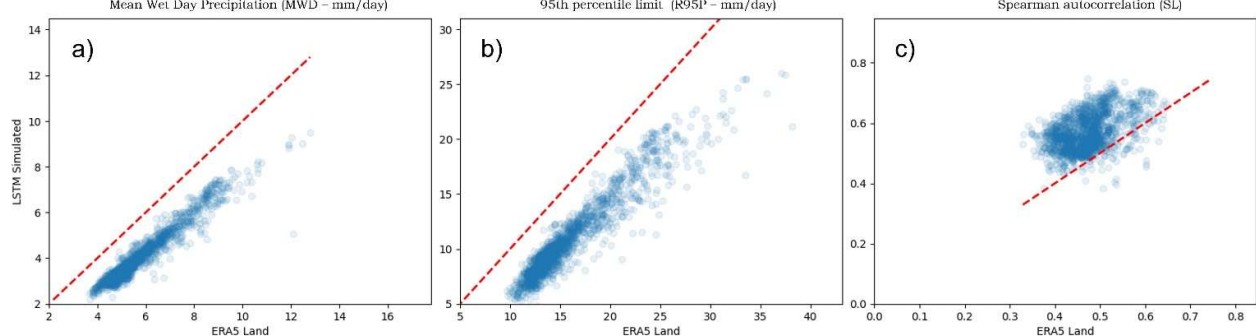

**Figure 4 Scatterplots for the three timeseries measures a) mean wet day precipitation (MWD) – mm/day, b) 95th percentile limit (R95P) – mm/day, and c) Spearman autocorrelation values (SL) between ERA5 Land and *with_discharge* LSTM Simulated. Each point represents a single catchment within the dataset. A 1:1 line (shown as a red dotted line in Fig) indicates overestimation/underestimation bias.**

**3.3 Out of sample predictions**

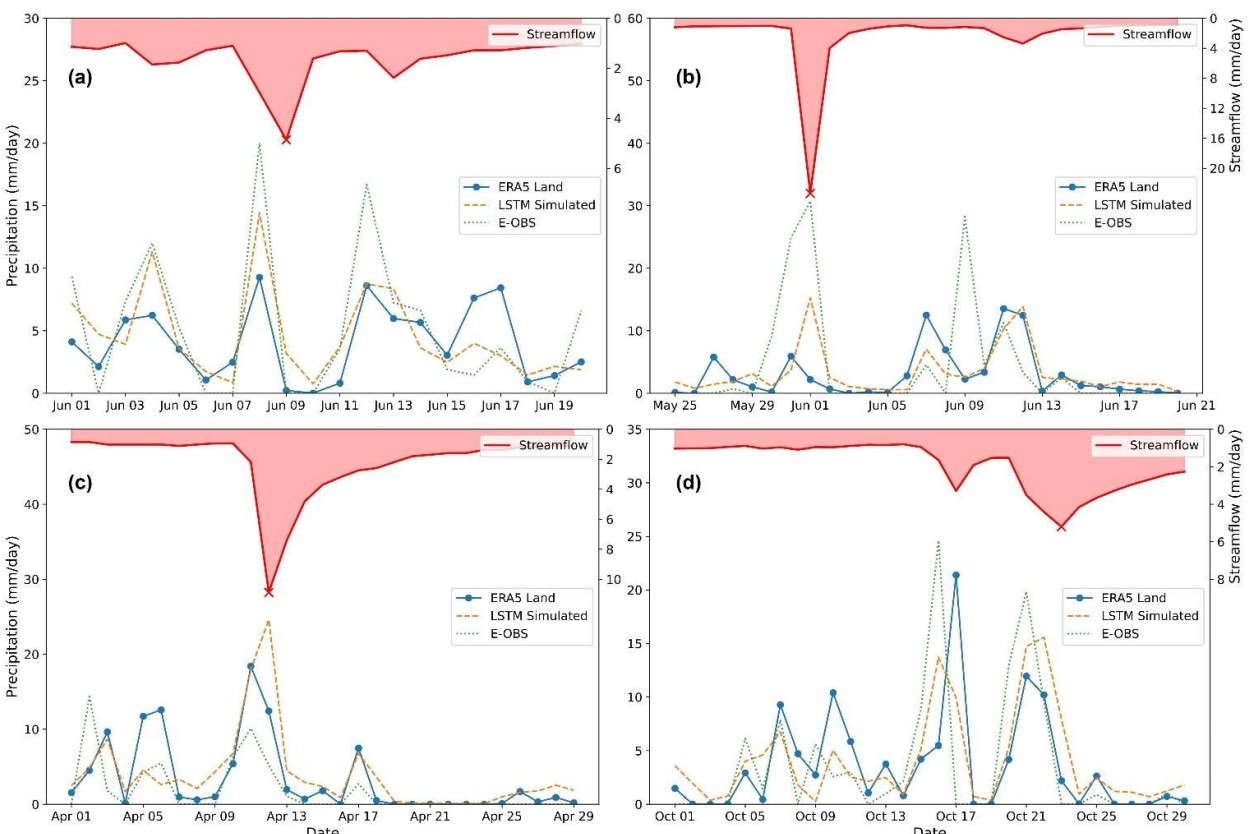

**Figure 5 Precipitation estimates for flood events at the four out of sample catchments: (a) – Elsenz Schwarzbach, (b) – Ernz, (c) –**
**Sueiro, (d) – Hoelzlebruck. The red line indicates the observed daily streamflow (with the day of the flood indicated by a cross). The**
**orange curve denotes the precipitation amount predicted by the *with_discharge* LSTM model, while the blue line depicts the original**
**ERA5 Land time series, and the green line indicates the estimate from the gauge based E-OBS product.**

Figure 5 shows predicted event precipitation values over time for the four out-of-sample catchments. Again, we compare the inversely modelled values (*with_discharge*) to the original ERA5 Land (used for training) and the gauge-based E-OBS product.
Table 3 lists the peak storm precipitation values reported by the different products along with the recorded flood values (both normalised to the catchment area in mm/day). Also shown are the storm runoff coefficients for the respective events based on the different precipitation estimates and discharge data.

Figure 5A represents the summer flood in June 2016 in the Elsenz Schwarzbach catchment in Germany. This annual flood event was triggered by a series of convective rainfall events caused by persistent atmospheric conditions in Germany during
the summer of 2016. Localised rainfall totals exceeded 100 mm in some catchments (Bronstert et al., 2018), triggering widespread flash floods. Our previous work (Manoj J et al., 2024) indicated that ERA5 Land could not accurately replicate the characteristics of the convective storm that caused this annual flood event over the Elsenz Schwarzbach catchment. A comparison of *with_discharge* LSTM-simulated precipitation values revealed estimates closer to those reported in the observational E-OBS product. When comparing with the E-OBS, the relative underestimation error in precipitation reduced
from around 100% (ERA5 Land) to 40% (*with_discharge*). The runoff coefficient for the event also decreased from 35% (ERA5 Land) to around 23% (*with_discharge*), which is consistent with estimates from Manoj et al. (2024).

Next, the *with_discharge* model was used to estimate precipitation for another convective episode over the Ernz Catchment in Luxembourg (Fig. 5B) in the summer of 2018. There was a noticeable improvement in the precipitation time series for both timing and peak storm values compared to ERA5 Land. While ERA5 Land completely missed this storm, the *with_discharge*
model was able to represent the sharp rise and descent of the curve. However, the runoff coefficients and peak storm values

(Table 2) indicate that the *with_discharge* LSTM model underestimates the true precipitation amount. In the third catchment (Sueiro: camelses_1414 from Caravan Spain extension), the *with_discharge* estimate for storm forcing was higher than ERA5 Land and E-OBS (Fig. 5C). The corresponding runoff coefficients underline the reliability of the storm prediction from *with_discharge* (0.37) compared to E-OBS (1.05).

In the Hoelzlebruck catchment (camelsch_4003 from Caravan Switzerland extension), two consecutive events occurred in October 2014. ERA5 Land was better than the *with_discharge* LSTM model in capturing the initial event magnitude, while the *with_discharge* model had better timing accuracy for the events. For the second event, which was the annual flood event, the *with_discharge* model, which incorporated streamflow information, was again able to reduce the relative errors in precipitation magnitudes (Fig. 5D)

**Table 3 Event characteristics for the four out of sample catchments**

| Event Characteristics | | Elsenz-Schwarbach | Ernz | Sueiro | Hoelzlebruck |
|---|---|---|---|---|---|
| **Precipitation (mm/day)** | ERA5 Land | 10.62 | 9.15 | 39.8 | 28.55 |
| | *with_discharge* | 16.45 | 23.49 | 64.83 | 44.68 |
| | E-OBS | 20.03 | 49.43 | 22.54 | 42.33 |
| **Discharge (mm/day)** | | 3.75 | 27.12 | 23.68 | 20.85 |
| **Runoff Coefficient (-)** | ERA5 Land | 0.35 | 2.96 | .60 | 0.73 |
| | *with_discharge* | 0.23 | 1.15 | .37 | 0.47 |
| | E-OBS | 0.19 | 0.55 | 1.05 | 0.49 |

To determine if the out-of-sample catchment performance could be solely attributed to discharge information, we utilized the *without_discharge* model (which only received meteorological forcings) to inversely predict the forcing precipitation for the 2016 flood event in the Elsenz Schwarzbach. Figure 6 shows that the *without_discharge* model was unable to capture the

driving storm dynamics as effectively as the *with_discharge* model and, therefore, could not accurately rectify the estimates from ERA5 Land.

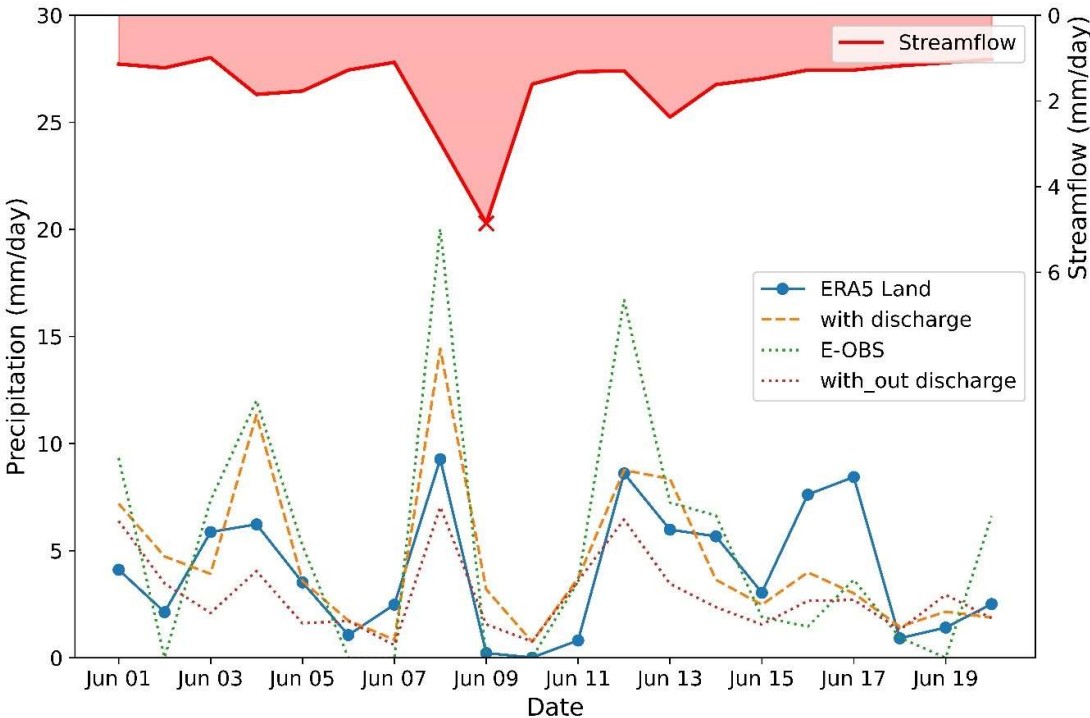

**Figure 6 Precipitation estimates for the flood event on June 8, 2016, at the Elsenz Schwarzbach. The red line represents the observed daily streamflow, with a cross marking the day of the flood. The orange curve illustrates the precipitation amount predicted by the with_discharge LSTM model, while the dotted red line represents the without_discharge model. The blue line depicts the original ERA5 Land time series, and the green line shows the estimate from the gauge-based E-OBS product.**

### 3.4 Forward Hydrological Modelling

The precipitation estimates generated by the *with_discharge* LSTM model were then used to run classical hydrological models (HBV and CATFLOW: Table B1) in a forward manner. To address the question of performance in differently sized basins, we run the conceptual HBV model in two catchments (Fig. S5) - Elsenz Schwarzbach (Fig. 7: 196.5 km$^2$) and Lippe (Fig. 8: 3366.3 km$^2$).

Figure 7 illustrates that the HBV model, which utilized the inverted precipitation estimates, performed slightly better (NSE = 0.64) during the evaluation period over Elsenz Schwarzbach compared to the model driven by the ERA5 Land (NSE = 0.57). To gain a better understanding of the differences between the models, we visually examined the results for three individual flood events, as shown in Fig. 7A-C.

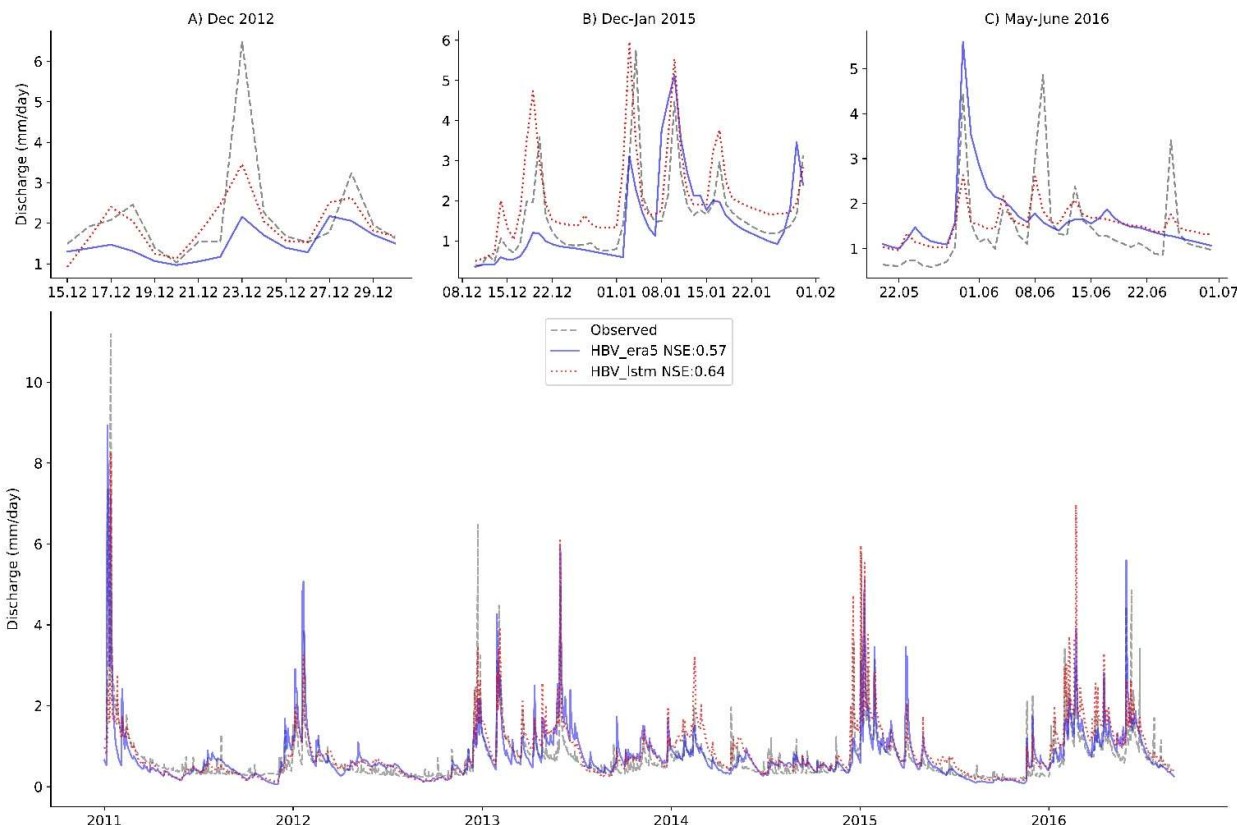

**Figure 7 Observed and simulated runoff (using the HBV model) at the Elsenz Schwarzbach catchment. The blue line denotes the streamflow simulated using the ERA5 Land precipitation product, while the red curve depicts the simulations using the inversely-estimate precipitation obtained using the *with_discharge* LSTM model. Moreover, three rainfall-runoff events are highlighted and displayed separately.**

During the winter flood of December 2012 (23 Dec 2012, Fig. 7A), the model driven by ERA5 Land significantly underestimated both the peak and the volume of the flood event. When using *with_discharge* -simulated precipitation, the relative peak error decreased by nearly 25%. Similarly, the model runs using *with_discharge* precipitation more accurately captured the pre-event conditions (18 Dec 2012) and the post-event conditions (28 Dec 2012). This aligns with findings from other studies (Berghuijs et al., 2019; Manoj J et al., 2023) that emphasize the importance of initial conditions for floods across Europe.

In the winter of 2015 (Fig. 7B), the model using *with_discharge* precipitation once again demonstrated better performance (albeit with overestimation errors). During the convective summer storm event in 2016 (Fig. 7C), neither model run successfully captured the flashy runoff response. Although the model that utilized ERA5 Land input predicted an earlier flood event in May 2016 with an overestimation bias, it did not accurately depict the dynamics of the annual flood event occurring a few days later. In contrast, the model with LSTM-generated precipitation (*with_discharge*) generally performed better in capturing both the magnitude and timing of the smaller storm peaks as well as the annual flood event on June 8, 2016.

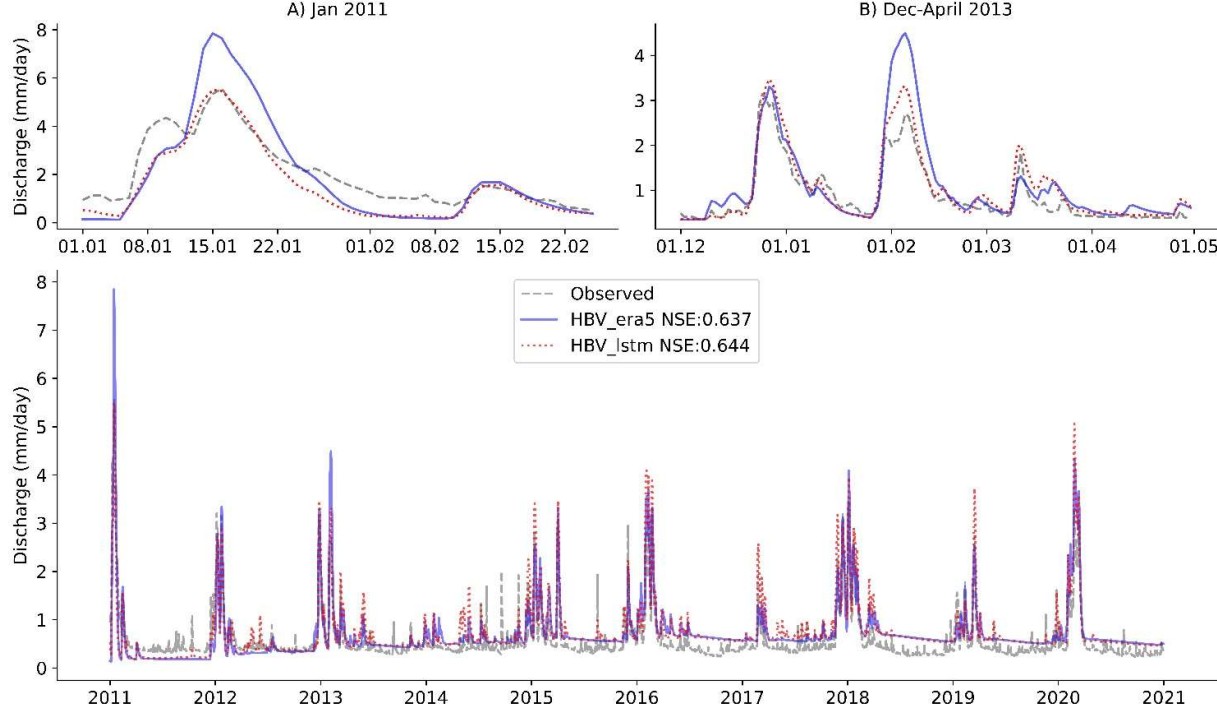

**Figure 8 Observed and simulated runoff (using the HBV model) at the Lippe catchment. The blue line denotes the streamflow simulated using the ERA5 Land precipitation product, while the red curve depicts the simulations using the inversely-estimate precipitation obtained using the *with_discharge* LSTM model. Moreover, two rainfall-runoff events are highlighted and displayed separately.**

For the larger Lippe catchment, we again saw comparable mean performance for both the runs (Fig. 8). For the winter flood of 2011 (Fig. 8A), the HBV model, which used inversely generated precipitation, closely matched the observed streamflow dynamics, whereas the ERA5 Land run exhibited significant overestimation errors. The inversely generated precipitation estimates again improved HBV model performance for replicating the discharge dynamics during the floods in December 2012 and February 2013 (Fig. 8B).

To understand the evolution of soil moisture dynamics while using the *with_discharge* LSTM-based precipitation estimates in physically based models, we conducted a hillslope-scale CATFLOW model simulation (Loritz et al., 2017; Manoj J et al., 2024) in one of the headwater catchments in Elsenz Schwarzbach (ERA5 Land vs *with_discharge* LSTM). The pairwise correlation values, as shown in Fig. 9, indicate that the use of the LSTM-based precipitation estimates does not lead to a loss of information regarding soil moisture dynamics in the region. In fact, we observe a slight increase in correlation when comparing the inversely derived precipitation estimates (referred to as CATFLOW_lstm) to MERRA and GLDAS (Table 1), in contrast with the correlation obtained for the run with ERA5 Land (referred to as CATFLOW_era5). As expected, the correlation value for the ERA5 Land run is slightly higher when assessed against soil moisture from the same ERA5 Land dataset, which may be attributed to model biases arising from using the same dataset for both precipitation and soil moisture.

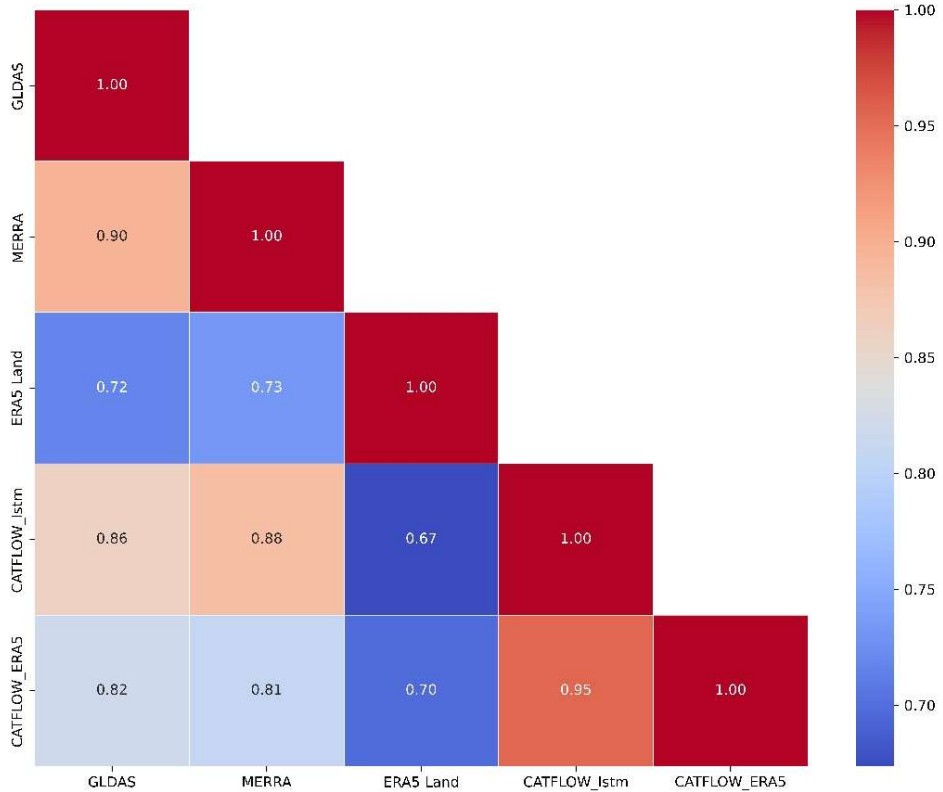

**Figure 9 Correlation matrix plot illustrating the pairwise correlations between the different soil moisture estimates - GLDAS (NASA Global Land Data Assimilation System, GLDAS-2.2 GRACE DA: Li et al., 2019), MERRA (Modern-Era Retrospective analysis for Research and Applications version 2 – tavg1_2d_lnd_Nx: Global Modeling And Assimilation Office, 2015), ERA5 Land: (Muñoz-Sabater et al., 2021), CATFLOW_lstm: model run using inversely estimated precipitation estimate from the LSTM model and CATFLOW_ERA5: model run using precipitation estimate from ERA5 Land product.**

## 4 Discussion

### 4.1 Improved precipitation estimation using discharge

Overall, our study reiterates that streamflow data can be exploited to obtain useful information about the nature of catchment-scale effective precipitation: we can thus invert the cause using the effect as input to an LSTM. This is in line with, and steps beyond, previous studies (Brocca et al., 2013; Kirchner, 2009; Kretzschmar et al., 2014; Krier et al., 2012; Teuling et al., 2010) that explored the possibility of doing hydrology backwards using experimental catchments. Here, we successfully expanded this idea to large samples, cutting across the wide range of hydro-climatic conditions that characterise Europe. We found a largely 'normal' distribution of performance, with a few outliers, the latter indicating possible poor quality of discharge data.

Although ERA5 Land precipitation has known uncertainties, it provides continuous global spatial and temporal coverage, making it a useful training dataset. Our goal was not to generate a fully independent dataset but to improve the ERA5 Land precipitation estimates using the additional streamflow information. Reanalysis data, by definition, are a mix of observations and past short-range weather forecasts rerun with modern weather forecasting models. Different data assimilation methods are then employed (Li et al., 2019). The inversion technique could be used as another final layer of post-processing (using the LSTM in this case) for the model outputs to ensure that the final product is more consistent with the variabilities observed in the discharge record.

One limitation of our approach is that the LSTM model tends to underestimate the timeseries measures (MWD and R95P) at the continental scale. The LSTM's architecture is known to have a theoretical saturation limit, leading to the underestimation

of some of the peak storm events. This so called 'saturation problem' (Baste et al., 2025; Chen and Chang, 1996) implies that irrespective of the input series, the predicted values can never exceed a theoretical limit (which is established during the training phase). Furthermore, the LSTM model looks for recurrence in patterns and mean conditions. This means that it can indeed account for consistent baseflow dynamics (as also indicated by analysis over the larger Lippe catchment, Figure 8). In extreme floods (Merz et al., 2021), the relative contributions of each component can vary significantly, depending on various factors

such as the antecedent conditions of the catchment area. The model likely struggles to learn this variability while attempting to invert and obtain the driving precipitation values. Given the non-linear nature of the inverse problem, there are always multiple possible solutions. Since the model is trained to minimize the mean squared error (Gupta et al., 2009), it may also tend to consistently predict lower values (on peaks) to effectively reduce the average error during training.

It is also important to acknowledge that 'true' precipitation estimates don't exist at the catchment scale. We obtain estimates
of forcing precipitation at such scales (with considerable uncertainty) by interpolating station data (e.g. E-OBS) or averaging gridded data from reanalysis/remote sensing products (e.g. ERA5 Land). Studies evaluating daily precipitation from EOBS and ERA5 over Europe (Bandhauer et al., 2022) have shown that while E-OBS is superior to ERA5 in regions with dense data, using ERA5 has advantages in data-scarce regions. The same was true for out of sample analysis (Fig. 5). For the Sueiro catchment (camelses_1414), the closest observational station is located more than 60 km away (Fig. S4), this explains why the
EOBS performs rather poorly in representing the driving forcings for the summer flood event (Fig. 5C).

The performance comparison using the rain gauge based EOBS product was intended to provide insight into the feasibility of different precipitation estimates from a hydrological perspective. While we acknowledge the existence of even better regional products (e.g., HYRAS – German Weather Service) for some of the study catchments, we believe that these various products should not be viewed as independent of one another. Instead, they contain complementary information as they represent the
same physical truth i.e. precipitation occurring over a catchment, albeit with different uncertainties and errors.

## 4.2 Catchment as a functional unit

In the introduction, we argued that the catchment scale is crucial for improving our understanding of the factors that drive the water cycle and representing them more accurately in reanalysis products. Our findings across the four catchments highlight the benefit of using streamflow variations to rectify precipitation estimates. By leveraging the generalisation capabilities of the
data-driven LSTM model, we successfully transferred knowledge across different scales (Notably, only about 9% of the catchments in our training dataset had areas smaller than 100 km²), indicating important implications for addressing the ever-evolving challenge of predictions in ungauged basins (PUB: Hrachowitz et al., 2013)

Although this approach can only be applied after the event has taken place, it has implications for generating coherent long-term statistical records for catchment forcings, which could be used for the design of small- to medium-purpose water resource
projects. Employing daily precipitation sums from products like ERA5 Land and EOBS should ideally be a last resort for reproducing small-scale hydrological events, however, the scarcity of real-world data and the rarity of these events may sometimes necessitate a modelling decision to incorporate these coarser estimates. Using the streamflow fluctuations, it would be possible to identify localised rainfall cells or snowfall events that are poorly captured by traditional rain gauges (Kretzschmar et al., 2014). The approach also has potential for evaluating long-term rainfall estimates from Global Circulation
Models for specific catchments using information about hydrological conditions (Fujihara et al., 2008).

While the LSTM-based precipitation estimates improved the representation of most events, there were still instances where the original ERA5 Land provided better accuracy for peak flood magnitudes (Fig. 5); this highlights the need for a blended approach that incorporates additional information rather than completely replacing one product with another. In regions around the world, the wealth of streamflow information remains underutilised in this aspect. For Germany alone (Loritz et al., 2024a),

there are more than 1500 streamflow gauges, which represent a significantly higher representative area compared to precipitation stations.

The forward exercise using the HBV model showed that the precipitation estimates after inversion enhanced mean performance for streamflow simulation and helped improve the modelling of extreme individual floods. The ability to match the hydrograph differed between the different seasons. Compared to the storage-controlled winter floods (Dunne and Kirkby, 1978) , summer floods in these regions are usually driven by Hortonian flow (Horton, 1932) in response to high-intensity rainfall during convective storms. Previous studies (Kirchner, 2009; Krier et al., 2012) have discussed such storage-controlled dynamics and their impact on the inversion problem.

Previous experiences at the event scale (Beauchamp et al., 2013; Zehe and Blöschl, 2004) have also shown that inferring the antecedent soil moisture conditions remains a key challenge for accurate and reliable flood simulations. By utilising the process-based CATFLOW model for soil moisture simulations in a small headwater catchment, we achieved high correlation values using the inverse precipitation estimate. This suggests that the approach can help represent the catchment's overall water dynamics and has the potential for reliable flood design estimations at the event scale, particularly in data-scarce regions.

**4.3 Limitations and Outlook**

It is important to stress that, as for any data-driven study, the results of our work are contingent on the quality of the training dataset. While we are aware of better regional products for individual countries, ERA5 Land provides consistent global coverage, and a permissive data sharing policy makes it one of the obvious choices for a continental scale modelling exercise. To evaluate the applicability of the commonly used LSTM network architecture, we decided to use the same architecture previously employed in hydrological studies instead of creating an experimental design with modified individual layers and training functions for inverse modelling. It is evident that exploring the impacts of different loss functions and deep learning model architectures like transformers would help advance the methodology discussed in this paper. This approach could also shed light on best-suited algorithms for the problem but is beyond the scope of the present work. The choice of Mean Squared Error (MSE) as the training function and Nash Sutcliffe Efficiency (NSE) as a performance metric is motivated by its success and applications in the forward problem (streamflow prediction), but this adds its own biases to the modelling exercise. In the present work, we tried to overcome this issue by relying less on the evaluation measure (NSE) and placing greater emphasis on the hydrological feasibility of the predictions (using the runoff coefficient). Additionally, we tried to complement this by calculating various other time series metrics commonly used in hydrometeorological studies. The four events for out-of-sample tests across various catchments were chosen based on the severity of the floods and ERA5 Land's inability to capture the characteristics of the driving storms. The choice of the hydrological models and calibration period also adds uncertainty to the forward simulations.

Our approach opens up many perspectives for future research. Transfer learning to data-scarce regions could help address the challenge of highly uncertain precipitation estimates in smaller catchments without precipitation gauges, improving hydrological modeling and the representation of extreme events such as convective storms, which are crucial for designing flood defense measures. Additionally, the inversion technique could serve as a final post-processing layer for gridded reanalysis products, ensuring better consistency with discharge variability and enabling machine learning approaches to estimate spatial precipitation fields conditioned on discharge data (Bárdossy et al., 2022, 2020). Moreover, this methodology could be applied to reconstruct past floods by leveraging historical hydrological records, storm water level markings, and observational flood data (Bronstert et al., 2018; Seidel et al., 2009), providing valuable insights into the driving storms behind some of the devastating past flood events. The workflow could also be expanded for the generation of new precipitation products, merging multiple different precipitation sources alongside the streamflow inversion.

## 5 Conclusions

Our main hypothesis was supported by the findings, which demonstrated that discharge has unused potential and can be inversely assimilated to adjust precipitation estimates derived from reanalysis products, while machine learning models are key to expanding this effort to large data sets spanning the scale of entire continents. The continental-scale analysis revealed that while the characteristics for the various time series attributes are well represented at the continental scale, there remain

significant underestimation biases compared to the original reanalysis product. Insights from the out-of-sample catchments provided valuable information about the applicability of our method for estimating flood forcings and the generalizability of the model. Additionally, we have shown that the inversely estimated precipitation estimates can improve forward modelling of both streamflow and soil moisture dynamics, illustrating how the information gained can be integrated into existing modelling strategies.

**Appendix A: LSTM configurations**

Table A1 details the static and dynamic inputs used for setting up the *with_discharge* and *without_discharge* LSTM models. The hyperparameter settings for both models are shown in Table A2, while Figure A1 provides the comparison results for both runs.

**Table A1 Model configurations for the LSTM model runs.**

| Model | Inputs | | Output |
|---|---|---|---|
| | **Static Attributes** | **Dynamic Attributes** | |
| *with_discharge* | *area* (area of catchment – km$^2$) <br> *p_mean* (mean daily precipitation – mm/day) <br> *ele_mt_sav* (spatial mean elevation – m above sea level) <br> *frac_snow* (fraction of precipitation falling as snow) <br> *pet_mm_syr* (potential evapotranspiration annual mean - mm) | *temperature_2m_mean* (daily mean temperature - °C) <br> *surface_net_solar_radiation_mean* (shortwave radiation – Wm$^{-2}$) <br> *surface_net_thermal_radiation_mean* (Net thermal radiation at the surface - Wm$^{-2}$) <br> *qobs_lead* (lead streamflow 7 days – mm/day) | *total_precipitation_sum* (precipitation daily sums – mm/day) |
| *without_discharge* | *area* (area of catchment – km$^2$) <br> *p_mean* (mean daily precipitation – mm/day) <br> *ele_mt_sav* (spatial mean elevation – m above sea level) <br> *frac_snow* (fraction of precipitation falling as snow) <br> *pet_mm_syr* (potential evapotranspiration annual mean - mm) | *temperature_2m_mean* (daily mean temperature - °C) <br> *surface_net_solar_radiation_mean* (shortwave radiation – Wm$^{-2}$) <br> *surface_net_thermal_radiation_mean* (Net thermal radiation at the surface - Wm$^{-2}$) | *total_precipitation_sum* (precipitation daily sums – mm/day) |

**Table A2 Hyperparameter settings for the LSTM models**

| Hyperparameter | LSTM Network |
|---|---|
| Hidden Layer | 1 |
| Hidden cells | 64 |
| Batch size | 256 |
| Sequence length | 365 |
| Epochs | 5 |
| Drop out | 0.4 |
| Learning rate | 0.001 |
| Optimizer | Adam |

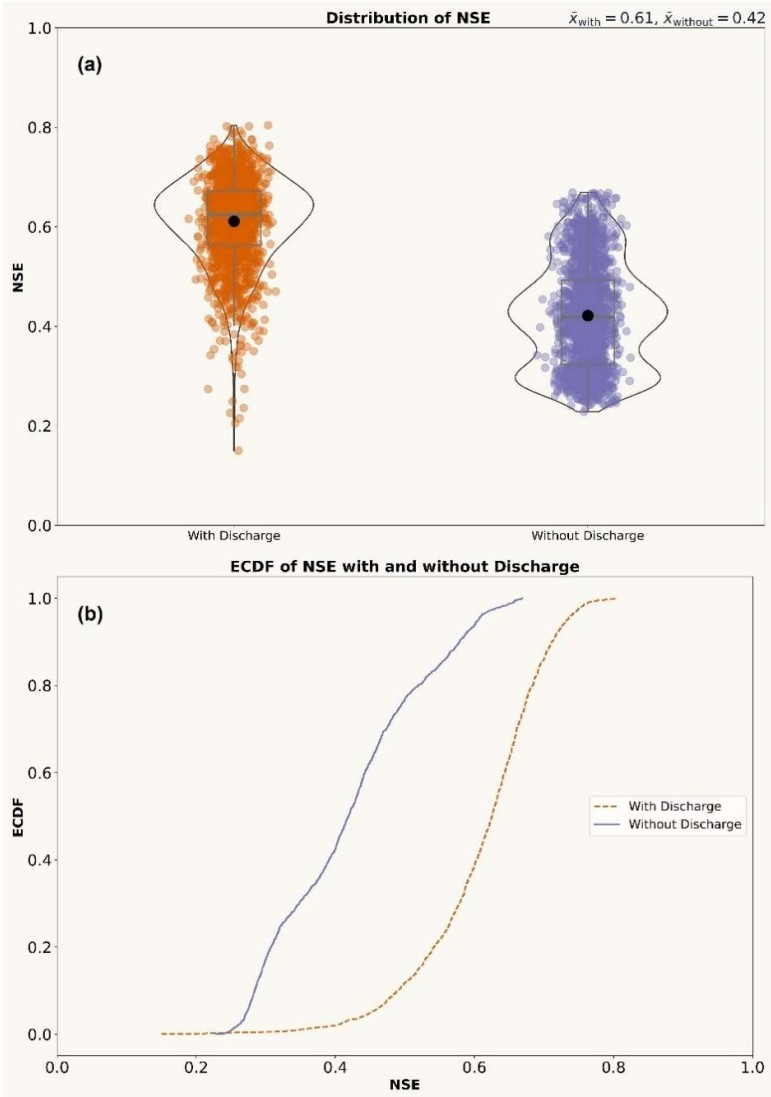

**Figure A1 Comparison of the mean performance of the two regional scale LSTM models (*with_discharge* and *without_discharge*). (a) Top panel depicts violin plots with included boxplots showing the distribution of performance (quantified by comparing the LSTM model simulated precipitation series to the original ERA5 Land timeseries over the testing period: NSE) (b) Bottom panel displays cumulative distribution plots for the performance of the two models.**

 **Appendix B: Hydrological Modelling**

**Hydrologiska Byråns Vattenbalansavdelning (HBV)**: The HBV model (Bergström and Forsman, 1973) is a so-called conceptual hydrological model that is used to simulate rainfall-runoff processes at the catchment scale. It makes use of different catchment water stores (storage elements, also referred to as buckets). Each storage element represents a certain compartment

of a catchment (e.g. groundwater, surface water bodies, soil zone). The main input requirements include precipitation, temperature and potential evapotranspiration. The model has several empirical parameters that need to be calibrated during the model training phase. A more detailed description of the model architecture and set up can be found in the studies by Seibert (2005) and Loritz et al. (2024a).

**CATFLOW:** The physically based model CATFLOW for catchment water and solute dynamics was developed as part of the detailed process studies carried out from 1991 – 1996 in the Weiherbach catchment in South-West Germany (Zehe et al., 2001). The basic modeling unit is a 2-D hillslope, discretized by curvilinear orthogonal coordinates in the vertical and downslope directions. Soil water dynamics within the hillslopes are characterized using the potential based form of the 2D Darcy–Richards equation. Overland flow is simulated using the diffusion wave approximation of the Saint-Venant equation

and explicit upstreaming, in combination with the Gauckler-Manning-Strickler formula. A detailed model description with the workflow required for setting up the model can be found in Manoj J et al. (2024).

**Table B1 Validation test cases for the hydrological models**

| Model | HBV (Bergström and Forsman, 1973) | | | | CATFLOW (Zehe et al., 2001) | |
|---|---|---|---|---|---|---|
| **Catchment (Figure S5)** | Elsenz Schwarzbach (Manoj J et al., 2024) | | Lippe (Loritz et al., 2024a) | | Headwater catchment W32 in Elsenz Schwarzbach (Manoj J et al., 2024) | |
| **Area (km²)** | 196.5 | | 3366.3 | | 5.6 | |
| **Scenario** | Using original ERA5 Land precipitation | Using inversely generated precipitation (*with_discharge*) | Using original ERA5 Land precipitation | Using inversely generated precipitation (*with_discharge*) | Using original ERA5 Land precipitation | Using inversely generated precipitation (*with_discharge*) |
| **Forcings** | ERA5 Land precipitation, potential evapotranspiration (Hargreaves), air temperature (ERA5 Land) | Inversely generated precipitation, potential evapotranspiration (Hargreaves), air temperature (ERA5 Land) | ERA5 Land precipitation, potential evapotranspiration (Hargreaves), air temperature (ERA5 Land) | Inversely generated precipitation, potential evapotranspiration (Hargreaves), air temperature (ERA5 Land) | ERA5 Land precipitation, potential evapotranspiration (ERA5 Land), plant and soil parameters from Manoj J et al. (2024) | Inversely generated precipitation, potential evapotranspiration (ERA5 Land), plant and soil parameters from Manoj J et al. (2024) |
| **Calibration period** | 01.01.2000 – 31.12.2010 | | 01.01.1990 – 31.12.2010 | | Uncalibrated predictions | |
| **Validation period** | 01.01.2011 – 31.12.2016 | | 01.01.2011 – 31.12.2020 | | 01.01.2008 – 31.12.2015 | |
| **Outputs compared** | Streamflow | | Streamflow | | Soil moisture | |
| **NSE** | 0.57 | 0.64 | 0.637 | 0.644 | 0.70 – 0.82 | 0.67-0.88 |

## Appendix C: Performance Metrics

**Nash-Sutcliffe Efficiency (NSE)** - First proposed by Nash and Sutcliffe (1970), the Nash–Sutcliffe efficiency (NSE) is one of the most widely used similarity measures in hydrology for calibration, model comparison, and verification. It measures how well the simulated timeseries ($y_{sim}$) matches the observed values ($y_{obs}$).

$$NSE = 1 - \frac{\sum(y_{obs} - y_{sim})^2}{\sum(y_{obs} - \bar{y}_{obs})^2} \qquad \text{Eqn. C1}$$

Values closer to 1 indicate excellent model performance (D. N. Moriasi et al., 2007), while NSE values near or below 0 suggest that the model, in fact, performs worse than simply using the mean of the observed values.

**Mean Wet Day Precipitation** (**MWD**: mm/day) – The Expert Team on Climate Change Detection and Indices (ETCCDI - World Climate Research program; 2021) recommends evaluating the intensity of precipitation on wet days (defined as a day with a minimum of 1 mm precipitation) to understand systematic over or underestimation of precipitation amounts. This metric (Simple Daily Intensity Index as per ETCCDI) is reported as the mean daily precipitation on days where precipitation > 1 mm. Let $P_i$ be the daily precipitation amount on wet days, ($P_i > 1mm$). If $N$ represents the total number of wet days, then:

$$MWD = \frac{\sum_{i=1}^{N} P_i}{N} \qquad \text{Eqn. C2}$$

**95th Percentile Precipitation** (**R95P**: mm/day) – This metric denotes the daily precipitation value at which 95% of all daily values (again only considering rainy days) are lower (top 5% events). This helps to assess the ability to capture extreme precipitation events. Let $P_i$ be the daily precipitation amount on wet days, ($P_i > 1mm$).

$$R95P = Percentile\ (\{P_i | P_i > 1mm\}, 95) \qquad \text{Eqn. C3}$$

**Spearman Rank Autocorrelation (SL)** - The Spearman Rank Autocorrelation measures the monotonic relationship between daily precipitation values and their values on the preceding day (1-day lag). It is computed using the ranked values of the precipitation time series. For a precipitation timeseries (with total $n$ observations) $P = \{P_1, P_2, ..., P_n\}$ with $R(P_i)$ and $R(P_{i+1})$ being the ranks of the precipitation values at times $t$ and $t + 1$,

$$SL = 1 - \frac{6\sum_{t=1}^{n-1}\left(R(P_{t+1}) - R(P_t)\right)^2}{n(n^2 - 1)} \qquad \text{Eqn. C4}$$

This measure helps analyse persistence in precipitation patterns and whether the temporal structure of precipitation events are preserved.

## Code Availability

The codes used to conduct the LSTM analysis in this study are based on the publicly available HY$^2$DL python library (https://github.com/KIT-HYD/Hy2DL) and can be accessed at https://doi.org/10.5281/zenodo.15051718 (Manoj J, 2025a). The code used to run the HBV models is available at https://doi.org/10.5281/zenodo.15051981 (Manoj J, 2025b). The CATFLOW model and the setup used to run the experiment in this study are archived at https://doi.org/10.5281/zenodo.10958813 (Manoj J, 2024) .

## Data Availability

The Caravan dataset and related community extensions are publicly available at https://doi.org/10.5281/zenodo.10968468 (Kratzert et al., 2023) and https://github.com/kratzert/Caravan/discussions/10. We acknowledge the E-OBS dataset from the Copernicus Climate Change Service (C3S, https://surfobs.climate.copernicus.eu) and the data providers in the ECA&D project (https://www.ecad.eu). The datasets generated as part of this publication can be found at https://doi.org/10.5281/zenodo.15051718 (Manoj J, 2025a) and https://doi.org/10.5281/zenodo.15051981 (Manoj J, 2025b).

## Author Contribution

AMJ designed the study and carried out all analysis and model simulations. Funding was acquired by EZ. The initial draft was prepared by AMJ, with all authors contributing to review and editing. RL, HG and EZ jointly supervised the work. All authors have read and agreed to the current version of the paper.

## Competing interests

At least one of the (co-)authors is a member of the editorial board of Hydrology and Earth System Sciences.

## Acknowledgements

The authors acknowledge support by the federal state of Baden-Württemberg through bwHPC (High Performance Computing Cluster). AMJ would like to thank Eduardo Acuña Espinoza for helpful discussions regarding the HY$^2$DL python library for deep learning methods.

## Financial Support

AMJ would like to thank the German Research Foundation (DFG) for financial support (Implementation of an InfraStructure for dAta-BasEd Learning in environmental sciences: ISABEL - 496155047).

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
