# Peer review of "Can discharge be used to inversely correct precipitation?"

_Hydrology and Earth System Sciences, 2024_

## Referee Comment (RC1)

**Review hess-2024-375**

**General comments:**

The authors present a method to improve the estimation of catchment-average effective precipitation from the ERA5 product by utilizing the information contained in stream flow data and a regional LSTM model.

To validate this interesting approach the authors model the runoff using this catchment-average effective precipitation as forcing and compare it to the runoff in the CAMELS data set. Averaged over all catchments contained in the CAMELS data set this approach improves the modelled runoff compared to using only ERA5 precipitation estimates as forcing.

The paper is well written and has a reasonable length. However, I think the authors could address the topic of "scale" more in depth. This starts at describing the used data sets in more detail, especially by mentioning their spatial and temporal resolution. Furthermore, the selection of the out-of-sample data sets as "proof-of-concept" is restricted to very small-scale basins. It is especially at this scale that daily ERA5 precipitation will most likely not perform well as forcing for a hydrological model due to its coarse spatial resolution.

The authors state that the LSTM model is estimating catchment-average precipitation amounts. This implies that the introduced approach might not perform equally well for differently-sized basins, when the runoff dynamics shift from surface-runoff to baseflow dominated basins. In my opinion the authors should elaborate more on this topic.

The study is interesting and introduces a promising approach to improve gridded precipitation products which is why I recommend its publication in NHESS after addressing the following comments.

**Specific comments:**

- The question about the feasibility of the introduced approach (as stated in the abstract) could be answered more clearly. While the authors show that their approach indeed improves the modelled discharge, they could elaborate more on how this method should be actually applied. E.g. a general improvement of products like ERA5: Does the method have potential to adjust the precipitation estimates of ERA5, e.g. as a pre-processing step? What could be general use-cases of the method?

- The authors state that the method could be useful to "reproduce small-scale high-impact events" (l.15-16) which also reflects in their choice of out-of-sample catchments. However, in my opinion it should be stated more clearly (e.g. in the

"Limitations") that using daily precipitation sums of products like ERA5 and EOBS should really be the last resort when trying to reproduce small-scale hydrological events (with or without LSTM correction). The selected flood events in four out-of-sample catchments were dominated by sub-daily rainfall and a rapid, probably also sub-daily, flood response that can just not be captured with daily sums and the spatial and temporal resolution of ERA5 and EOBS.

- In Figure 3 you are comparing ERA5 and EOBS precipitation estimates to the effective precipitation generated by the LSTM. Therefore, the lower amounts from the LSTM seem obvious. How useful is this comparison between two different types of precipitation?

- **Line 318**: *"this consistent underestimation also seems physically plausible"*. Do I understand correctly that the LSTM-model systematically underestimates the precipitation in order to generate less surface runoff, to compensate for lacking baseflow dynamics? The systematic underestimation of precipitation from the LSTM model (as seen in Fig. 3) seems problematic to me because then the inversely derived precipitation would be only useful to model the discharge in surface-runoff dominated catchments. The four out-of-sample catchments are all quite small: How would the LSTM-derived precipitation perform when forcing a hydrological model in a larger out-of-sample catchment that is dominated by base-flow dynamics? Please clarify this, because in my view this is a key point. You could also consider splitting the results of the two regional LSTM models into groups (by basin size) to investigate whether the underestimation of precipitation is more problematic for larger catchments when modelling the discharge.

- If you use "catchment-average effective precipitation" as forcing for HBV and CatFlow, what happens with the soil water dynamics, and interception? Do they get subtracted again from the effective precipitation input? Maybe the overall wording is misleading and the LSTM is not really returning effective precipitation?

- The authors mention that their approach could be useful in "data-scarce regions world wide". However, the uneven distribution or lack of stream gauges could also introduce a bias to the LSTM model.

- A very brief explanation of the used data sets in the data section would be helpful for the reader, even though most of these data sets are commonly used. The temporal and spatial resolution of these data sets has to be mentioned, as well as the way these data sets are derived (e.g. based on station data). The

authors do this partially, e.g., in line 128, for the E-OBS dataset but I would recommend doing it in a more structured way, for instance in a table.

- CARAVAN
- ERA5
- E-OBS
- CAMELS

You could also consider mentioning MERRA and GLDAS already here. Additionally, you could consider splitting the "Data and Methods" section into "Data" and "Methods" to provide more clarity.

- Please describe your measures of goodness briefly and why chose to use them: mean wet days, spearman lag, 95th percentile. For mean wet days you should also mention the unit mm, otherwise this gets confusing. This also applies to Figure 3 where the unit "mm" is missing

- **Line 121**: Please write out the abbreviations for the five catchment static attributes.

- **Line 96**: "the model was provided with a 7-day lead time series for discharge" Don't you also have to provide the 7-day lead time series of the other forcings? Can you explain why you decided for a lead time of 7 days? Is the lead time not catchment specific/scale dependent, or does 7 days cover everything?

**Technical corrections:**

Generally, the plots are not displayed nicely in the PDF. I am not sure if the reason is the processing by HESS or if you should increase the resolution.

**Line 40**: The reference "*Clerc-schwarzenbach et al.,2024*" should be written with a capital "S": Clerc-Schwarzenbach

**Line 45:** The reference Manoj J et al. is missing a "." after "J".

**Line 46**: "*precipitation forcings data*". This sentence seems a bit awkward. How about. "precipitation forcing data"?

**Line 50**: "*it is usually the rainstorm events occurring in poorly observed areas that lead to high impacts*": This sentence could be misunderstood in a way that high-impact-rainstorms preferably happen in areas because they are poorly observed,

while in fact the observation network is too sparse and the majority of high-impact-rainstorms is simply not observed.

**Line 56-59**: *"While the classical "forward rainfall-runoff generation problem" has received considerable attention over various decades (Montanari et al., 2013; Sivapalan et al., 2003), a smaller subset of studies (Brocca et al., 2013; Kirchner, 2009; Kretzschmar et al., 2014; Krier et al., 2012; Teuling et al., 2010) has investigated the feasibility of tackling the inverse problem more efficiently."*

What do you mean with "more efficiently"? More efficient than what?

**Line 59**: Reference in the wrong format: Kirchner (Kirchner, 2009) should be Kirchner (2009).

**Line 81**: You could already make a reference to Table 2, to reveal more information about the out-of-sample catchments.

**Line 90**: References in wrong format. Parenthesis should just be around the year. E.g. Kratzert et al. (2018).

**Line 93**: Reference in wrong format. Should be "Loritz et al., 2024".

**Line 104 and 105**: Why are you using two different references for the HBV model?

**Line 156**: You already spelled out the abbreviation of the HBV model previously and do not need to do it again here.

**Line 184**: "underlying causes of precipitation". What do you mean by this? Maybe "causes" is the wrong word?

**Line 190**: I think you confuse "without_discharge" and "with_discharge" here. It looks like "with_discharge" has a steeper curve.

**Figure 3:** Please add the units, otherwise it is unclear whether it is [days] or [mm] for "mean (wet days)". You mention "mean wet days" before but explain the unit first in line 204.

**Line 241: "***precipitation values revealed closer estimates to those reported***"** -> I think it should be: precipitation values revealed estimates closer to those reported.

**Table 2**: The first column of the table is not correctly formatted.

**Line 326-334:** This paragraph seems unnecessarily long. If your main message is that products like ERA5 have a too coarse resolution to capture small-scale precipitation events you can shorten this. Instead, you could elaborate here on the potential of your method, improving such products.

**Line 374**: *"adds its own biases to the modelling exercise"* Can you explain this?

**Figure S2 and S3:** Minor detail: Maybe it would look better, if you put the north-arrow in some corner, so it is less prominent? In Figure 3 there is no legend, so you could at least mention the red triangles in the caption.

**Figure S5:** How did you derive the gridded precipitation? By interpolating the rain gauges? It is not recommended using the rainbow color map because it is not perceptually uniform and therefore not easy to distinguish for people with some kind of color blindness. See also the HESS guidelines, section "color schemes": https://www.hydrology-and-earth-system-sciences.net/submission.html#figurestables

---

## Author Comment (AC1)

**Can discharge be used to inversely correct precipitation? (hess-2024-375)**

Ashish Manoj J, Ralf Loritz, Hoshin Gupta, Erwin Zehe

The authors would like to thank Anonymous Reviewer 1 for carefully reviewing our manuscript and for providing their valuable comments and suggestions, which we believe will be very helpful in improving the overall structure and quality of the manuscript. The following responses have been prepared to address all the reviewers' comments point-by-point. We have responded (in black) to the reviewer's comment (in blue).

**General comments:**

The authors present a method to improve the estimation of catchment-average effective precipitation from the ERA5 product by utilizing the information contained in stream flow data and a regional LSTM model.

To validate this interesting approach the authors model the runoff using this catchment-average effective precipitation as forcing and compare it to the runoff in the CAMELS data set. Averaged over all catchments contained in the CAMELS data set this approach improves the modelled runoff compared to using only ERA5 precipitation estimates as forcing.

The paper is well written and has a reasonable length. However, I think the authors could address the topic of "scale" more in depth. This starts at describing the used data sets in more detail, especially by mentioning their spatial and temporal resolution. Furthermore, the selection of the out-of-sample data sets as "proof-of-concept" is restricted to very small-scale basins. It is especially at this scale that daily ERA5 precipitation will most likely not perform well as forcing for a hydrological model due to its coarse spatial resolution.

The authors state that the LSTM model is estimating catchment-average precipitation amounts. This implies that the introduced approach might not perform equally well for differently-sized basins, when the runoff dynamics shift from surface-runoff to baseflow dominated basins. In my opinion the authors should elaborate more on this topic.

The study is interesting and introduces a promising approach to improve gridded precipitation products which is why I recommend its publication in NHESS after addressing the following comments.

We thank the reviewer for constructive, supportive suggestions and for highlighting the work's potential. We have prepared the following points to address the main comments raised by the reviewer:

A. Although the out-of-sample catchments we selected are relatively small, our approach using the LSTM model—trained on much larger catchments—showed skill at adjusting the (under) estimated precipitation values for these events at such smaller scales. This is noteworthy given that ERA5 Land precipitation typically performs poorly at this scale; only about 9% of the catchments in our training dataset had areas smaller than 100 km². This finding demonstrates the ability of our inversion methodology and spatial transfer learning (He et al., 2011) to effectively leverage knowledge from data-rich, well-represented regions to make predictions in data-scarce areas.

B. To address the question of performance in differently sized basins, we will supplement the existing analysis by an evaluation of the methodology over larger, previously unseen test catchments from the recently published Caravan CAMELS-DE (Loritz et al., 2024) dataset. The results (Figure 1: camelsde_DEA11130) again indicate a reduction in relative errors in peak discharge and flood volume for such large catchments (>3000 km²), using the inversely generated precipitation.

C. We agree with the reviewer that our usage of the term "effective precipitation" requires more clarity. To begin with, it is important to stress that precipitation uncertainty is rarely considered when quantifying model output uncertainty; while studies are usually conducted to show how differences in simulated discharge can be as a consequence of changing precipitation input, they rarely look at how much improvement of the model performance would be possible by using different but plausible precipitation (Bárdossy et al., 2022, 2020). "True" precipitation estimates are not known at the catchment scale. We obtain estimates of them (with considerable uncertainty) by either interpolating station data or averaging gridded data from reanalysis/remote sensing products. Our aim was to generate a precipitation time series (estimate) that is more "consistent" with the dynamics captured in the discharge record. We then benchmarked our new estimate by using forward hydrological models over the Elsenz Schwarzbach (Figures 6 & 7 in the original draft) and Lippe catchment (Figure 1 in this

document). As our results indicate, the inversely generated precipitation estimate reduces both volume and peak errors for the HBV model simulation in both catchments.

We will update the manuscript to reflect these points and include the new analysis.

[Figure]

Figure 1 Observed and simulated runoff (using the HBV model) at the Lippe catchment (camelsde_DEA11130). The blue line denotes the streamflow simulated using the ERA5 Land precipitation product, while the red curve depicts the simulations using the inversely-estimate precipitation obtained using the regional LSTM model. Moreover, two rainfall-runoff events are highlighted and displayed.

**Specific comments:**

- The question about the feasibility of the introduced approach (as stated in the abstract) could be answered more clearly. While the authors show that their approach indeed improves the modelled discharge, they could elaborate more on how this method should be actually applied. E.g. a general improvement of products like ERA5: Does the method have potential to adjust the precipitation

estimates of ERA5, e.g. as a pre-processing step? What could be general use-cases of the method?

We agree with the reviewer that more information about the proposed method's application could enhance the work's potential. Some of the main points we will add are:

A. Transfer learning to data-scarce regions: For a number of smaller catchments, no precipitation gauges exist within the catchment boundaries. Due to this, highly uncertain and erroneous precipitation estimates are often employed for hydrological modelling over these catchments, leading to the underrepresentation of high impact events such as convective storms. The inverse methodology could be applied to generate more realistic representations of extreme precipitation statistics at these scales, which could then be utilised to design flood defence measures.

B. Improvement of gridded products: Reanalysis data, by definition, are a mix of observations and past short-range weather forecasts rerun with modern weather forecasting models (ECMWF, 2023). The inversion technique could be used as another final layer of post-processing for the model outputs to ensure that the final product is more consistent with the variabilities observed in the discharge record. In line with this, it would also be interesting to use machine learning and other data-driven approaches to generate estimates of the spatial fields of precipitation (Bárdossy et al., 2022, 2020) conditioned on the discharge information.

C. Reconstruction of past floods: As a more general use case, the methodology could be applied to reconstruct information about the driving storms that caused some of the devastating past floods. There exists a wealth of hydrological information about such events (Bronstert et al., 2018; Seidel et al., 2009), either in the form of storm water level markings or observational flood records.

- The authors state that the method could be useful to "reproduce small-scale high-impact events" (l.15-16) which also reflects in their choice of out-of-sample catchments. However, in my opinion it should be stated more clearly (e.g. in the "Limitations") that using daily precipitation sums of products like ERA5 and EOBS should really be the last resort when trying to reproduce small-scale hydrological events (with or without LSTM correction). The selected flood events in four out-of-sample catchments were dominated by sub-daily rainfall and a rapid, probably also sub-daily, flood response that can just not be captured with daily sums and the spatial and temporal resolution of ERA5 and EOBS.

The reviewer has raised a well-justified concern about using coarse-resolution precipitation products to reproduce small-scale hydrological events. While it is not ideal to use only such products, the scarcity of real-world data and the rarity of these events sometimes necessitate a modelling decision to incorporate these coarser estimates. We will address this as described in our response to the main comment.

- In Figure 3 you are comparing ERA5 and EOBS precipitation estimates to the effective precipitation generated by the LSTM. Therefore, the lower amounts from the LSTM seem obvious. How useful is this comparison between two different types of precipitation?

True precipitation rates at the catchment scale are seldom accurately known. The various products used carry significant uncertainty, particularly when applied in rainfall-runoff models. The main goal of our analysis at the continental spatial scale was to highlight systemic biases and identify areas where further investigation could help address these discrepancies. We found that while the LSTM model consistently underestimated extreme values, the spatial gradients were still well represented.

- **Line 318**: *"this consistent underestimation also seems physically plausible"*. Do I understand correctly that the LSTM-model systematically underestimates the precipitation in order to generate less surface runoff, to compensate for lacking baseflow dynamics? The systematic underestimation of precipitation from the LSTM model (as seen in Fig. 3) seems problematic to me because then the inversely derived precipitation would be only useful to model the discharge in

surface-runoff dominated catchments. The four out-of-sample catchments are all quite small: How would the LSTM-derived precipitation perform when forcing a hydrological model in a larger out-of-sample catchment that is dominated by base-flow dynamics? Please clarify this, because in my view this is a key point. You could also consider splitting the results of the two regional LSTM models into groups (by basin size) to investigate whether the underestimation of precipitation is more problematic for larger catchments when modelling the discharge.

- If you use "catchment-average effective precipitation" as forcing for HBV and CatFlow, what happens with the soil water dynamics, and interception? Do they get subtracted again from the effective precipitation input? Maybe the overall wording is misleading and the LSTM is not really returning effective precipitation?

We thank the reviewer for demanding more clarity about the consistent underestimation by the LSTM model. We feel that this can be attributed to the following reasons:

a) The LSTM model looks for recurrence in patterns and mean conditions. This means that it can indeed account for consistent baseflow dynamics (as also indicated by our new analysis over the larger Lippe catchment, Figure 1). In extreme floods (Merz et al., 2021), the relative contributions of each component can vary significantly, depending on various factors such as the antecedent conditions of the catchment area. The model likely struggles to learn this variability while attempting to invert and obtain the driving precipitation values.

b) Given the non-linear nature of the inverse problem, there are always multiple possible solutions. Since the model is trained to minimize the mean squared error (Gupta et al., 2009), it may also tend to consistently predict lower values (on peaks) to effectively reduce the average error during training.

c) Recent studies have shown that the LSTM models have a theoretical upper limit for prediction based on weights and biases of the head linear layer (Espinoza et al., 2024; Kratzert et al., 2024). In simpler terms, irrespective of the input series, the predicted values can never exceed a theoretical limit

(which is established during the training phase). This so called 'saturation problem' (Chen and Chang, 1996; Rakitianskaia and Engelbrecht, 2015) of the LSTM architecture would also lead to the underestimation of some of the peak storm events.

We will discuss all these relevant points and opt for clearer wording concerning the precipitation estimate from the LSTM model in the revised version of the manuscript.

- The authors mention that their approach could be useful in "data-scarce regions world wide". However, the uneven distribution or lack of stream gauges could also introduce a bias to the LSTM model.

This is a valid point as data scarcity is a pertinent challenge in hydrological modelling, however, as shown in the IAHS PUB (Prediction in Ungauged Basins: Sivapalan et al., 2003) decade, there is much knowledge to be gained from spatial transfer learning. Concerning the LSTM model, we believe that while it will consistently perform better in regions similar to those it has already encountered during training, it also possesses some generalization capabilities. If these capabilities are implemented in a hydrologically sound way, they could significantly help address the ungauged basin problem.

- A very brief explanation of the used data sets in the data section would be helpful for the reader, even though most of these data sets are commonly used. The temporal and spatial resolution of these data sets has to be mentioned, as well as the way these data sets are derived (e.g. based on station data). The authors do this partially, e.g., in line 128, for the E-OBS dataset but I would recommend doing it in a more structured way, for instance in a table.

- CARAVAN
- ERA5
- E-OBS
- CAMELS

You could also consider mentioning MERRA and GLDAS already here. Additionally, you could consider splitting the "Data and Methods" section into "Data" and "Methods" to provide more clarity.

We will follow the reviewer's suggestion to have a dedicated Data section with a description of the different products. A new table detailing the spatial and temporal resolution of all the data sources used in the study will also be added.

- Please describe your measures of goodness briefly and why chose to use them: mean wet days, spearman lag, 95th percentile. For mean wet days you should also mention the unit mm, otherwise this gets confusing. This also applies to Figure 3 where the unit "mm" is missing

The goodness of fit measures will be explained in the Methodology section. The figure 3 will also be updated to show the units.

- **Line 121**: Please write out the abbreviations for the five catchment static attributes.

This will be added to the revised manuscript.

- **Line 96**: "the model was provided with a 7-day lead time series for discharge"

  Don't you also have to provide the 7-day lead time series of the other forcings? Can you explain why you decided for a lead time of 7 days? Is the lead time not catchment specific/scale dependent, or does 7 days cover everything?

The 7-day lead time was chosen as this provided a reasonable upper estimate for the time of concentration in the catchments. Since the objective was not on forecasting, we opted not to give the lead time values for the other meteorological variables. While the lead time values are indeed catchment-specific, since the catchment area is already given as a static attribute to the LSTM model, we expect the model to learn this dependency from the data.

**Technical corrections:**

Generally, the plots are not displayed nicely in the PDF. I am not sure if the reason is the processing by HESS or if you should increase the resolution.

We can confirm that we have thoroughly rechecked the quality of each individual figure. The decline in PDF quality is likely due to the conversion of embedded figures from the original Word document. This issue will be resolved during the final typesetting phase, as we will provide the original high-resolution images to HESS. We agree with all the technical corrections proposed by the reviewer and will incorporate the necessary changes into the revised draft.

**Line 40**: The reference "*Clerc-schwarzenbach et al.,2024*" should be written with a capital "S": Clerc-Schwarzenbach

Thank you for pointing this out. We will correct the reference in the revised version.

**Line 45:** The reference Manoj J et al. is missing a "." after "J".

This will be added.

**Line 46**: "*precipitation forcings data*". This sentence seems a bit awkward. How about. "precipitation forcing data"?

We will change this to *precipitation forcing data.*

**Line 50**: "*it is usually the rainstorm events occurring in poorly observed areas that lead to high impacts*": This sentence could be misunderstood in a way that high-impact-rainstorms preferably happen in areas because they are poorly observed, while in fact the observation network is too sparse and the majority of high-impact-rainstorms is simply not observed.

We appreciate the reviewer's point about the need for more clarity. This sentence will be rephrased to avoid any ambiguity.

**Line 56-59**: "*While the classical "forward rainfall-runoff generation problem" has received considerable attention over various decades (Montanari et al., 2013; Sivapalan et al., 2003), a smaller subset of studies (Brocca et al., 2013; Kirchner, 2009; Kretzschmar et al., 2014;*

*Krier et al., 2012; Teuling et al., 2010) has investigated the feasibility of tackling the inverse problem more efficiently."*

What do you mean with "more efficiently"? More efficient than what?

Thank you for pointing out this typo. The term 'more' will be omitted in the new manuscript.

**Line 59**: Reference in the wrong format: Kirchner (Kirchner, 2009) should be Kirchner (2009).

We will correct this in the revised draft.

**Line 81**: You could already make a reference to Table 2, to reveal more information about the out-of-sample catchments.

This will be added to the revised version.

**Line 90**: References in wrong format. Parenthesis should just be around the year. E.g. Kratzert et al. (2018).

**Line 93**: Reference in wrong format. Should be "Loritz et al., 2024".

The references will be corrected.

**Line 104 and 105**: Why are you using two different references for the HBV model?

The first reference is the more general publication introducing the HBV model, while the second refers to the specific version of HBV used in our study. We will standardise the references throughout the revised draft.

**Line 156**: You already spelled out the abbreviation of the HBV model previously and do not need to do it again here.

We will omit the repetition.

**Line 184**: "underlying causes of precipitation". What do you mean by this? Maybe "causes" is the wrong word?

This will be changed to 'driving precipitation' in the revised draft.

**Line 190**: I think you confuse "without_discharge" and "with_discharge" here. It looks like "with_discharge" has a steeper curve.

We agree with the reviewer on the ambiguity of the statement. This will be removed in the revised draft.

**Figure 3:** Please add the units, otherwise it is unclear whether it is [days] or [mm] for "mean (wet days)". You mention "mean wet days" before but explain the unit first in line 204.

We thank the reviewer for pointing out the inconsistency in our usage. The same will be corrected, and units will be added in the revised version.

**Line 241: "***precipitation values revealed closer estimates to those reported***" -> I think it should be: precipitation values revealed estimates closer to those reported.**

We will update this to *precipitation values revealed estimates closer to those reported.*

**Table 2**: The first column of the table is not correctly formatted.

The table will be reformatted in the revised draft.

**Line 326-334:** This paragraph seems unnecessarily long. If your main message is that products like ERA5 have a too coarse resolution to capture small-scale precipitation events you can shorten this. Instead, you could elaborate here on the potential of your method, improving such products.

We will completely rewrite this paragraph, adding information about the application of the proposed method and its potential to improve gridded products.

**Line 374**: "*adds its own biases to the modelling exercise*" Can you explain this?

Our goal was to emphasize that in data-driven models, the choice of training function and evaluation metric significantly influences the results, especially when we focus solely on the numerical values of the goodness of fit measure. In this study, we aimed to address this issue by relying less on the evaluation measure (NSE) and placing greater emphasis on the feasibility of our predictions through runoff coefficient analysis at the event scale.

We plan to discuss more on this in the revised version and will also include relevant references.

**Figure S2 and S3:** Minor detail: Maybe it would look better, if you put the north-arrow in some corner, so it is less prominent? In Figure 3 there is no legend, so you could at least mention the red triangles in the caption.

We will update both figures (to also show the new out-of-sample catchment -Lippe) and move the north arrow. The captions of Figure S3 will also be updated.

**Figure S5:** How did you derive the gridded precipitation? By interpolating the rain gauges? It is not recommended using the rainbow color map because it is not perceptually uniform and therefore not easy to distinguish for people with some kind of color blindness. See also the HESS guidelines, section "color schemes":

https://www.hydrology-and-earth-system-sciences.net/submission.html#figurestables

The gridded precipitation depicted in the figure is from a radar product (Kachelmannwetter, 2023) operating over the Elsenz Schwarbach. The original figure was taken from an earlier publication (Manoj J et al., 2024). We will update this figure to also include the new test catchment area over the Lippe.

**References**

Bárdossy, A., Anwar, F., Seidel, J., 2020. Hydrological Modelling in Data Sparse Environment: Inverse Modelling of a Historical Flood Event. Water (Switzerland) 12. https://doi.org/10.3390/w12113242

Bárdossy, A., Kilsby, C., Birkinshaw, S., Wang, N., Anwar, F., 2022. Is Precipitation Responsible for the Most Hydrological Model Uncertainty? Front. Water 4, 1–17. https://doi.org/10.3389/frwa.2022.836554

Bronstert, A., Agarwal, A., Boessenkool, B., Crisologo, I., Fischer, M., Heistermann, M., Köhn-Reich, L., López-Tarazón, J.A., Moran, T., Ozturk, U., Reinhardt-Imjela, C., Wendi, D., 2018. Forensic hydro-meteorological analysis of an extreme flash flood: The 2016-05-29 event in Braunsbach, SW Germany. Sci. Total Environ. 630, 977–991. https://doi.org/10.1016/j.scitotenv.2018.02.241

Chen, C.T., Chang, W. Der, 1996. A feedforward neural network with function shape autotuning. Neural Networks 9, 627–641. https://doi.org/10.1016/0893-6080(96)00006-8

ECMWF, 2023. Factsheet on Reanalysis [WWW Document]. 2023. URL https://www.ecmwf.int/en/about/media-centre/focus/2023/fact-sheet-reanalysis

Espinoza, E.A., Loritz, R., Kratzert, F., Klotz, D., Gauch, M., Chaves, Á., Bäuerle, N., Ehret, U., 2024. Analyzing the generalization capabilities of hybrid hydrological models for extrapolation to extreme events 1–17.

Gupta, H. V., Kling, H., Yilmaz, K.K., Martinez, G.F., 2009. Decomposition of the mean squared error and NSE performance criteria: Implications for improving hydrological modelling. J. Hydrol. 377, 80–91. https://doi.org/10.1016/j.jhydrol.2009.08.003

He, Y., Bárdossy, A., Zehe, E., 2011. A review of regionalisation for continuous streamflow simulation. Hydrol. Earth Syst. Sci. 15, 3539–3553. https://doi.org/10.5194/hess-15-3539-2011

Kachelmannwetter, 2023. Kachelmannwetter [WWW Document]. URL https://kachelmannwetter.com/de

Kratzert, F., Gauch, M., Klotz, D., Nearing, G., 2024. HESS Opinions: Never train a Long Short-Term Memory (LSTM) network on a single basin. Hydrol. Earth Syst. Sci 28, 4187–4201.

Loritz, R., Dolich, A., Acuña Espinoza, E., Ebeling, P., Guse, B., Götte, J., Hassler, S., Hauffe, C., Heidbüchel, I., Kiesel, J., Mälicke, M., Müller-Thomy, H., Stölzle, M., Tarasova, L., 2024. CAMELS-DE: hydro-meteorological time series and attributes for 1555 catchments in Germany. Earth Syst. Dyn. Discuss. 1–30.

Manoj J, A., Loritz, R., Villinger, F., Mälicke, M., Koopaeidar, M., Göppert, H., Zehe, E., 2024. Toward Flash Flood Modeling Using Gradient Resolving Representative Hillslopes. Water Resour. Res. 60. https://doi.org/10.1029/2023WR036420

Merz, B., Blöschl, G., Vorogushyn, S., Dottori, F., Aerts, J.C.J.H., Bates, P., Bertola, M., Kemter, M., Kreibich, H., Lall, U., Macdonald, E., 2021. Causes, impacts and patterns of disastrous river floods. Nat. Rev. Earth Environ. 0123456789. https://doi.org/10.1038/s43017-021-00195-3

Rakitianskaia, A., Engelbrecht, A., 2015. Measuring saturation in neural networks. Proc. - 2015 IEEE Symp. Ser. Comput. Intell. SSCI 2015 1423–1430. https://doi.org/10.1109/SSCI.2015.202

Seidel, J., Imbery, F., Dostal, P., Sudhaus, D., Bürger, K., 2009. Potential of historical meteorological and hydrological data for the reconstruction of historical flood events-the example of the 1882 flood in southwest Germany. Nat. Hazards Earth Syst. Sci. 9, 175–183. https://doi.org/10.5194/nhess-9-175-2009

Sivapalan, M., Takeuchi, K., Franks, S.W., Gupta, V.K., Karambiri, H., Lakshmi, V., Liang, X., McDonnell, J.J., Mendiondo, E.M., O'Connell, P.E., Oki, T., Pomeroy, J.W., Schertzer, D., Uhlenbrook, S., Zehe, E., 2003. IAHS Decade on Predictions in Ungauged Basins (PUB), 2003-2012: Shaping an exciting future for the hydrological sciences. Hydrol. Sci. J. 48, 857–880. https://doi.org/10.1623/hysj.48.6.857.51421

---

## Author Comment (AC2)

**Can discharge be used to inversely correct precipitation? (hess-2024-375)**

Ashish Manoj J, Ralf Loritz, Hoshin Gupta, Erwin Zehe

The authors would like to thank Anonymous Reviewer 2 for carefully reviewing our manuscript and giving their insightful comments and overall positive feedback. The following responses have been prepared to address all the reviewers' comments point-by-point. We have responded (in black) to the reviewer's comment (in blue).

This paper illustrates "doing hydrology backwards" by developing an LSTM model to predict precipitation based on reanalysis products, given meteorological inputs and the added input of catchment discharge. The authors show that a model given discharge does a better job at predicting precipitation, indicating that discharge encodes significant information about recent precipitation beyond other meteorological forcings. They also find that while the LSTM underestimates precipitation totals relative to the ERA5 training dataset, it better reproduces events that are poorly captured by ERA5.

This is a very interesting paper and appropriate for this journal. It effectively shows that a machine learning approach can be used to improve uncertain precipitation forcings, especially for short time-scale events that are not well represented in reanalysis. With this, I have several comments listed below that could improve and clarify some aspects. As a note, I see some of these may overlap with the first reviewer who also made good points and authors have already responded.

We thank the reviewer for constructive, supportive suggestions and for highlighting the work's potential. We have prepared the following points to address the main comments raised by the reviewer.

**General comments:**

This study poses that an LSTM model that is trained to reproduce ERA5 precip can actually estimate precip better than the ERA5 product itself. This is based on the input of "future" discharge, which encodes observed precipitation events that are not typically well captured by ERA5. In this way, the LSTM could deviate from the ERA5 because (a) ERA5 is not capturing precip as it actually occurred or (b) the LSTM is not performing well. Unless I am mistaken it seems hard to disentangle these, and the observation gage-based "E-OBS" product seems important here and could be better described. For

example, Figure 2 shows that the LSTM "with discharge" better replicates ERA5 precipitation than the LSTM "without" – and it is assumed that this better replication is a good thing. Meanwhile later figures illustrate differences in ERA5, LSTM, and E-OBS regarding specific events, but the LSTM "without" discharge is dropped. In general, it seems useful if E-OBS, ERA5, and both LSTM estimates could be compared up front to more clearly establish differences between them, i.e. what is currently done just between the LSTM models and ERA5 in Figure 2. As far as E-OBS, a few more details on that data might be beneficial especially in the events selected for Figure 5 and associated discussion. For example, what is the proximity of a gage to the specific study catchments?

We agree with the reviewer that some additional information is required for a better understanding of the results. Some of the relevant points we will discuss are:

A. "True" precipitation estimates are not known at the catchment scale. We obtain estimates of them (with considerable uncertainty) by either interpolating station data (EOBS) or averaging gridded data from reanalysis/remote sensing products (ERA5 Land). Our aim was to generate a precipitation time series (estimate) that is more "consistent" with the dynamics captured in the discharge record.

B. The comparison between the models *with_discharge* and *without_discharge* had two primary objectives. First, we aimed to determine whether the discharge values contained useful information about the corresponding precipitation. Second, we sought to evaluate if the LSTM model could effectively capture this non-linear relationship. The *without_discharge* served as a benchmark for evaluating the information gained from including discharge data. To avoid redundancy, we chose not to include these runs in the spatial maps. In response to the reviewer's comment, we conducted another comparison (Figure 1) of the model *without_discharge* for one of the out-of-sample catchments and again found that its performance was inferior to the model that incorporated discharge information.

C. More information regarding the EOBS observational product for the out of sample test will be provided in the revised draft. Figure 2 shows the proximity of observational stations (used for deriving EOBS) to the four specific catchments considered in this study.

[Figure]

Figure 1 Precipitation estimates for the flood event on June 8, 2016, at the Elsenz Schwarzbach. The red line represents the observed daily streamflow, with a cross marking the day of the flood. The orange curve illustrates the precipitation amount predicted by the with_discharge LSTM model, while the dotted red line represents the without_discharge model. The blue line depicts the original ERA5 Land time series, and the green line shows the estimate from the gauge-based E-OBS product.

[Figure]

Figure 2 Spatial maps showing the proximity of observational stations (used for deriving the EOBS gridded product) to the four out of sample catchments considered in the present study.

I can imagine that this method might be more effective for smaller catchments, and less effective for very large catchments where the effect of P on Q is more lagged and smoothed. For a very large catchment, estimating a single time-series of P based on Q seems like it would be trying to "average" multiple ERA-5 or gage-based grid cells. Some details on the spatial characteristics of the study catchments might be useful here, especially relative to the scale of gridded precipitation forcing. For example, is there any meaningful trend in model behavior (for any model) with catchment area, or are all study catchments relatively smaller in scale than any precipitation input that would be used?

To address the question of performance in differently sized basins (also asked by Reviewer 1), we will supplement the existing analysis by an evaluation of the methodology over larger, previously unseen test catchments from the recently published Caravan CAMELS-DE (Loritz et al., 2024) dataset. The results (Figure 3: camelsde_DEA11130) again indicate a reduction in relative errors in peak discharge and flood volume for such large catchments (>3000 km$^2$), using the inversely generated precipitation.

[Figure]

Figure 3 Observed and simulated runoff (using the HBV model) at the Lippe catchment (camelsde_DEA11130). The blue line denotes the streamflow simulated using the ERA5 Land precipitation product, while the red curve depicts the simulations using the inversely-estimate precipitation obtained using the regional with_discharge LSTM model. Moreover, two rainfall-runoff events are highlighted and displayed.

Although the out-of-sample catchments we selected are relatively small, our approach using the LSTM model—trained on much larger catchments—showed skill at adjusting the (under) estimated precipitation values for these events at such smaller scales. This is noteworthy given that ERA5 Land precipitation typically performs poorly at this scale; only about 9% of the catchments in our training dataset had areas smaller than 100 km$^2$. This finding demonstrates the ability of our inversion methodology and spatial transfer learning (He et al., 2011) to effectively leverage knowledge from data-rich, well-represented regions to make predictions in data-scarce areas.

With all the different models and products, a table or two might be useful – for example listing the properties of precipitation datasets, models and references, flow data. This could be linked to Figure 1 which gives the flow of the study.

We thank the reviewer for his suggestion. A new Table 1 will be added to the Data section detailing the spatial and temporal resolution of all the data sources used in the study.

Figures: figure captions could all be expanded or improved. For example, Figure 1needs a more descriptive caption that addresses the content of each panel and the connections. As it is, it does not really describe the flow of the study and could be a lot more useful to the reader. Otherwise, figures with (a), (b) (c) should be more clearly labeled as such in the captions, and figures like Figure 7 with no panels should not have any references to panels (a), (b), (c), etc. It is also a bit hard to compare the spatial images in Figure 3 because of the grey shading in the top 2 figures but not in the E-OBS panels (so it would be nice if the same masking could be applied to all of these maps). Finally in figures and text it should be made specific that when "LSTM" is mentioned in text or a caption that it is specified as one model or the other ("with" or "without" discharge).

We will implement the necessary changes in the revised draft. The captions for Figure 1 will be expanded to include detailed methodological steps that link to different sections of the paper. We will also correct the captions for Figure 7. For Figure 3, the grey area indicates the regions where catchment data is not included in this study. The EOBS was presented in its full spatial extent, as it is an interpolated product. Additionally, we will clarify the use of the *with_discharge* and *without_discharge* models throughout the manuscript.

**References**

He, Y., Bárdossy, A., Zehe, E., 2011. A review of regionalisation for continuous streamflow simulation. Hydrol. Earth Syst. Sci. 15, 3539–3553. https://doi.org/10.5194/hess-15-3539-2011

Loritz, R., Dolich, A., Acuña Espinoza, E., Ebeling, P., Guse, B., Götte, J., Hassler, S., Hauffe, C., Heidbüchel, I., Kiesel, J., Mälicke, M., Müller-Thomy, H., Stölzle, M., Tarasova, L., 2024. CAMELS-DE: hydro-meteorological time series and attributes for 1555 catchments in Germany. Earth Syst. Dyn. Discuss. 1–30.

---

## Author Comment (AC3)

**Can discharge be used to inversely correct precipitation? (hess-2024-375)**

Ashish Manoj J, Ralf Loritz, Hoshin Gupta, Erwin Zehe

The authors would like to thank Anonymous Reviewer 3 for going through our manuscript and giving their critical comments and suggestions. The following responses have been prepared to address all the reviewers' comments point-by-point. We have responded (in black) to the reviewer's comment (in blue).

**General comments:**

The paper by Manoj et al. describes a method to estimate catchment scale precipitation from streamflow records using a machine learning approach. The paper is well-written, and its objective is highly significant in the context of hydrological sciences, where precipitation data remain scarce and critical to improve water resources modelling. Using streamflow as a predictor to estimate rainfall is not new, but it makes perfect sense as streamflow reflects recent rainfall history. The LSTM model is perfectly justified for this task, considering the high level of performance reached by this type of machine learning approach. We particularly appreciated exploring a large sample of catchments, which reinforces the author's conclusions. We were also impressed with the final validation exercise using different hydrological models.

Overall, this paper contains many valuable elements and a tremendous amount of work. However, it suffers from two fundamental flaws requiring a major revision before its acceptance for publication:

We thank the reviewer for highlighting the work's significance and summarizing the main strengths. We propose to make the following changes to address the well justified concerns raised by the reviewer.

Comment #1: The fundamental aim of the paper stated in the introduction is to generate better precipitation estimates compared to currently available reanalysis products such as ERA5-Land. The authors are clear about the issues of reanalysis products in various parts of the manuscript. For example, related to a particular flood event, they indicate: "Our previous work (Manoj J et al., 2024) indicated that ERA5 Land could not accurately replicate the characteristics of the convective storm that caused this annual flood event" (line 240). Consequently, we do not see the point in training an LSTM using ERA5-Land

precipitation as a target. The best we can expect from this approach is to generate rainfall series identical to ERA5-Land precipitation, which is known to be problematic.

We agree that the ERA5 Land has issues representing the driving precipitation estimates for specific event scales (Essou et al., 2016; Manoj J et al., 2024). As stated in the beginning of our abstract, our aim was to see whether the inverse data assimilation using streamflow information could be used to overcome at least some of these well documented deficiencies.

a) Our approach, utilizing a regional LSTM model trained on much larger catchments, demonstrated effectiveness in adjusting the underestimated precipitation values for these events at smaller (out of sample) scales. Notably, only about 9% of the catchments in our training dataset had areas smaller than 100 km$^2$. We could show that discharge response encodes sufficient information about the driving precipitation to correct ERA5 Land in the right direction.

b) Reanalysis data, by definition, are a mix of observations and past short-range weather forecasts rerun with modern weather forecasting models (ECMWF, 2023). Different data assimilation methods are used for this. Our idea was that the inversion technique could be used as another final layer of post-processing (using the LSTM in this case) for the model outputs to ensure that the final product is more consistent with the variabilities observed in the discharge record.

Furthermore, any performance comparison between LSTM outputs (trained on ERA5-Land precip) and original ERA5-Land precip using an independent dataset as a reference (E-OBS in this case) are logically flawed: the LSTM was not trained to reproduce anything else than ERA5, so any perceived "improvement" between its outputs and ERA5-Land precip when simulating an independent dataset (E-OBS in this case) is due to chance. Fortunately, the solution to this problem is simple: instead of ERA5-Land, the authors could set the LSTM training target to rainfall observation (i.e. E-OBS). The comparison between LSTM and ERA5-Land would become meaningful and clarify if precipitation estimation can be improved compared to using ERA5-land.

We would like to clarify that our goal was to enhance the ERA5 Land estimates by incorporating streamflow information along with only other meteorological forcings

from ERA5 Land, rather than generating a new precipitation product that uses again another precipitation as input.

c) It is important to emphasize that "true" precipitation estimates are only available at observational stations and not at the catchment scale. The performance comparison using EOBS and the runoff coefficient was intended to provide insight into the feasibility of different precipitation estimates from a hydrological perspective. While we acknowledge the existence of even better regional products (e.g., HYRAS – German Weather Service) for some of the study catchments, we believe that these various products should not be viewed as independent of one another. Instead, they contain complementary information as they represent the same physical truth i.e. precipitation occurring over a catchment, albeit with different uncertainties and errors.

d) Studies evaluating daily precipitation from EOBS and ERA5 over Europe (Bandhauer et al., 2022) have shown that while E-OBS is superior to ERA5 in regions with dense data, using ERA5 has advantages in data scarce regions. This was also seen in the out of sample analysis. For the Sueiro catchment (camelses_1414), the closest observational station is located more than 60 km away (Figure 1 in this document), this explains why the EOBS performs rather poorly in representing the driving forcings for the summer flood event (Figure 5C in original draft). Additionally, the runoff coefficient estimate for E-OBS was around 1.05, which indicates a hydrologically infeasible value (Table 2 in the original draft) when compared to the estimates from ERA5 Land and LSTM. Compared to purely interpolated products like EOBS, reanalysis products are usually released and updated more frequently. This again points to the value of reanalysis products like ERA5 for tackling the *prediction in ungauged basin* (PUB - Hrachowitz et al., 2013) problem.

e) While our workflow could indeed be extended in multiple directions to generate more coherent precipitation products, we feel this is currently beyond the scope of our initial study, which aimed to explore whether discharge had sufficient information to help in tackling the inverse problem over a large sample space.

We will clarify these points and restructure the Introduction and Discussion sections to highlight them more effectively.

[Figure]

Figure 1 Spatial maps showing the proximity of observational stations (used for deriving the EOBS gridded product) to the four out of sample catchments considered in the present study.

Comment #2: When training their LSTM, the author used the mean catchment rainfall as a predictor (Pmean, see line 138). In other words, they use some of the predictand data as a static predictor. This is a major flaw in a regression setup: it gives the LSTM model a distinct advantage over an operational situation where, obviously, the mean catchment rainfall is not known. Here again, the solution to this issue is straightforward: remove this predictor from the list of static predictors.

While we used the LSTM model in the commonly used streamflow prediction (regression) mode, it is important to note that our end goal is different from an operational forecast. The model takes in future streamflow as a predictor, which implies that the real-time forecast implications of our methodology are limited. The approach could be seen as a data assimilation post-processing step to ensure that the final precipitation estimates are more consistent with the variabilities observed in the discharge record. We would like to highlight that hydrological modellers have previously

constrained their models (Gharari et al., 2021) using average annual runoff to improve calibration for the classical streamflow prediction problem.

To address the reviewer's comment regarding the removal of *p_mean* from the list of static attributes due to concerns about data leakage, we retrained and tested the regional scale LSTM model, removing *p_mean* while keeping all other conditions the same.

[Figure]

Figure 2 Empirical Cumulative Distribution Function (ECDF) of NSE values, comparing model performance in runs with and without *p_mean*.

The ECDF plot (Figure 2) indicate that the model performance remains fairly consistent for the two runs. We believe that this can be attributed to two main reasons:

A. Our training dataset had a large number of catchments (1804) cutting across various hydroclimatically diverse regions over Europe, ensuring that our model could learn robust dependencies from the meteorological (dynamic) forcings itself.

B. Recent research (Heudorfer et al., 2024; Li et al., 2022) on Entity Aware (EA) deep learning models (models that are provided with static features- predominantly in the form of physiographic proxies, next to dynamic forcing features) have suggested that the information in static features is not being effectively leveraged.

Li et al. (2022) demonstrated that an LSTM model using randomly initialized numbers as static features outperformed a model that used actual physiographic static features, as long as the number of random static features was greater than the number of physiographic static features. This indicates that the specific physiographic characteristics of the static features may be irrelevant; what truly matters is the presence of unique identifying information. Heudorfer et al. (2024) report that while static features serve as unique catchment identifiers, resulting in excellent in-sample performance when confronted with out-of-sample data, the model is unable to generalize from static features and instead relies almost exclusively on meteorological data for prediction.

Aside from these two fundamental problems, we also have a few general comments:

Comment #3: some aspects of the method lack clarity. We got a bit lost in all the cases considered by the authors at the end of the manuscript. We suggest clarifying several elements using summary tables in the method section (and not later in the paper):

- o the list of all LSTM configurations tested with their inputs (including lagged inputs) and their outputs,

- o the list of all performance metrics,

- o the list of all test cases including the number of catchments, the forcings (if using hydrological models) and the outputs tested.

We will include tables in the Data and Methods section that detail all the datasets used, as well as the various model runs and test cases. To enhance readability, we also plan to provide information about the LSTM configurations and the different hydrological models in a new appendix section.

Comment #3: The LSTM model was trained on mean squared error, emphasising large rainfall values. We recommend testing other configurations where training is done on transformed values, e.g. square roots and log transform, to check if certain rainfall metrics can be improved further.

We appreciate the reviewer's suggestion that testing additional functions could enhance some of the rainfall metrics. However, since the primary focus of our study was on accurately representing heavy precipitation events that lead to floods, we chose to use the mean square error training function because it was the simplest and most commonly used. We will include this information and suggest exploring other configurations as part of future research in this direction.

Detailed comments

Comment #4: Line 68, "we conjecture that the catchment-average precipitation can be inversely identified": this problem is still numerically ill-posed due to catchment memory. We suggest rephrasing to "we conjecture that streamflow data can reduce the uncertainty associated with this process by providing valuable information on recent rainfall history".

We will rephrase this line as suggested by the reviewer.

Comment #5: Line 115, "The Caravan dataset uses the ERA5 Land as meteorological forcing": it would be useful to remind that this is far from satisfactory as ERA5 is known to have important limitations when simulating rainfall.

This information will be added to the revised draft.

Comment #6: Line 185, "model "with_discharge" outperforms the model "without_discharge" not only on average but also concerning the best-performing catchments.": It would also be useful to show the distribution of pairwise NSE differences. This would answer the question: "How many catchments reach better NSE when using streamflow predictors?".

[Figure]

Figure 3 Violin plot displaying the pairwise differences (*with_discharge* vs *without_discharge* models) in NSE for the study catchments.

We agree with the reviewer's suggestion that displaying the distribution of pairwise NSE differences would better clarify our main results. Therefore, we will include pairwise NSE difference plots (Figure 3 in this document) in the main manuscript and move the distribution and ecdf plots (originally Figure 2 in the manuscript) to a new Appendix section dedicated to the LSTM models.

Comment #7: Figure 3: This map is a bit confusing because for LSTM and ERA5-Land, the data generated by the authors is in the form of points (i.e. catchment average), not surfaces. Please update the map accordingly.

Although we only have average information for the catchment area, these averages are derived from grid points that encompass the entire area. We believe that representing them as point data at the catchment outlets would not accurately reflect the fact that the information represents the whole catchment area.

Comment #8, Line 204 "preserves spatial gradients": what do the authors mean by "preserve"? Please clarify. It is hard to assess spatial patterns from maps as small as Figure 3. We suggest an additional metric and a figure to clarify this point.

We agree that it is indeed hard to assess the spatial gradients without additional metrics. Since evaluating such spatial gradients is not the main focus of the present study, we will remove this sentence from the revised draft.

Comment #9, source code: please list software requirements in the source code. This includes the list of software packages required and their versions. If the authors are using Anaconda, it can be done by adding to their repository a conda environment configuration file, also referred as "yml" file (conda contributors, 2025), which lists all Python package and their version.

A dependencies file detailing all the software packages and their versions will be added to the GitHub repository.

**References**

Bandhauer, M., Isotta, F., Lakatos, M., Lussana, C., Båserud, L., Izsák, B., Szentes, O., Tveito, O.E., Frei, C., 2022. Evaluation of daily precipitation analyses in E-OBS (v19.0e) and ERA5 by comparison to regional high-resolution datasets in European regions. Int. J. Climatol. 42, 727–747. https://doi.org/10.1002/joc.7269

ECMWF, 2023. Factsheet on Reanalysis [WWW Document]. 2023. URL https://www.ecmwf.int/en/about/media-centre/focus/2023/fact-sheet-reanalysis

Essou, G.R.C., Sabarly, F., Lucas-Picher, P., Brissette, F., Poulin, A., 2016. Can precipitation and temperature from meteorological reanalyses be used for hydrological modeling? J. Hydrometeorol. 17, 1929–1950. https://doi.org/10.1175/JHM-D-15-0138.1

Gharari, S., Gupta, H. V., Clark, M.P., Hrachowitz, M., Fenicia, F., Matgen, P., Savenije, H.H.G., 2021. Understanding the Information Content in the Hierarchy of Model Development Decisions: Learning From Data. Water Resour. Res. 57. https://doi.org/10.1029/2020WR027948

Heudorfer, B., Gupta, H., Loritz, R., 2024. Deep Learning Models in Hydrology Have Not Yet Achieved Entity Awareness. https://doi.org/10.22541/essoar.172927199.90156076/v1

Hrachowitz, M., Savenije, H.H.G., Blöschl, G., McDonnell, J.J., Sivapalan, M., Pomeroy, J.W., Arheimer, B., Blume, T., Clark, M.P., Ehret, U., Fenicia, F., Freer, J.E., Gelfan, A., Gupta, H. V., Hughes, D.A., Hut, R.W., Montanari, A., Pande, S., Tetzlaff, D., Troch, P.A., Uhlenbrook, S., Wagener, T., Winsemius, H.C., Woods, R.A., Zehe, E., Cudennec, C., 2013. A decade of Predictions in Ungauged Basins (PUB)-a review. Hydrol. Sci. J. 58, 1198–1255. https://doi.org/10.1080/02626667.2013.803183

Li, X., Khandelwal, A., Jia, X., Cutler, K., Ghosh, R., Renganathan, A., Xu, S., Tayal, K., Nieber, J., Duffy, C., Steinbach, M., Kumar, V., 2022. Regionalization in a Global Hydrologic Deep Learning Model: From Physical Descriptors to Random Vectors. Water Resour. Res. 58. https://doi.org/10.1029/2021WR031794

Manoj J, A., Loritz, R., Villinger, F., Mälicke, M., Koopaeidar, M., Göppert, H., Zehe, E., 2024. Toward Flash Flood Modeling Using Gradient Resolving Representative Hillslopes. Water Resour. Res. 60. https://doi.org/10.1029/2023WR036420

---

## Author Response (AR2)

**Can discharge be used to inversely correct precipitation? (hess-2024-375)**

Ashish Manoj J, Ralf Loritz, Hoshin Gupta, Erwin Zehe

Dear Dr. Roger Moussa,                                                    08.07.2025

Attached, please find the revised version of the manuscript "*Can discharge be used to inversely correct precipitation?*" co-authored with R. Loritz, H. Gupta and E. Zehe, to be considered for publication in Hydrology and Earth System Science.

After carefully reviewing the two comments from Anonymous Reviewer 3 in the previous round, we have decided to implement both the changes suggested by the reviewer. We downloaded and pre-processed the observational EOBS gridded data for all the catchments included in our training dataset. Next, we repeated the entire analysis using this observational product as the new training target. Additionally, we removed mean precipitation (*pmean*) from the list of static attributes in this updated setup. Lastly, we have made other minor changes facilitated by a fresh reading.

Our main finding remains largely unchanged in the updated version. We observed that incorporating discharge information improved the performance of the LSTM network during the unseen testing period and resulted in more hydrologically consistent storm estimates in the out-of-sample catchments. Additionally, the forward modelling using traditional hydrological models once again produced higher mean NSE values for the runs based on inversely generated precipitation estimates.

We would like to thank the Editor and Anonymous Reviewer 3 again for giving us another opportunity to revise our manuscript.

Please get in touch with me if you need any additional information.

Thank you very much for your consideration.

Best regards,

Ashish

On behalf of Ralf, Hoshin and Erwin

Email: ashish.jaseetha@kit.edu

**Reviewer 3:**

The authors would like to thank Anonymous Reviewer 3 for carefully reviewing our manuscript and providing their critical yet helpful and detailed comments. We have followed the reviewer's suggestions concerning both the comments. The following responses have been prepared to address all the reviewers' comments point-by-point. We have responded (in black) to the reviewer's comment (in blue).

**General comments:**

Following a first round of review, this review comments on a second version of the paper by Manoj et al., which describes a method to estimate catchment-scale precipitation from streamflow records using a machine learning approach. We reiterate positive comments we made in our first review, including how significant the paper is to practical hydrological problems and the use of large datasets to validate the results. We feel that the detailed comments indicated in our first review have been addressed satisfactorily by the authors. As a result, we won't mention them in this review.

Unfortunately, the authors have brushed aside our two main comments, despite the fact that we offered simple alternatives, and answered them with cosmetic changes in their manuscript as explained below. We respect their opinion, but cannot approve the publication of their paper under these conditions. As a result, we recommend a major revision of the paper.

The following sections clarify our position regarding the two fundamental points we raised in our first review.

Comment #1: Our first fundamental criticism of the paper is that it aims to generate an improved ERA5-Land precipitation product (referred to as ERA5-P hereafter) using a statistical model, but trains this model to reproduce ERA5-P. When the training is completed, the model being imperfect will introduce errors in its prediction, hence producing a contaminated version of ERA5-P (ERA5-P plus residual errors from the statistical model). The whole premise of the paper is to suggest that this contaminated version is significantly superior to the original ERA5-P. In other words, the residual error of the statistical model contains valuable information, even though it is being minimised by the training algorithm.

What we said in our first review is that this is possible, but it would be due to chance because the authors expect to find something valuable out of what the training algorithm discards. What is likely to happen is that this lucky outcome may be due to the author's focus on selected rainfall metrics. Other metrics may show that the contaminated ERA5-P is worse than the original ERA5-P, for example, rain event timing, long-term seasonality, sensitivity to warming climate conditions, zero rainfall simulations, etc… To summarise, what is lacking in the paper is a clear definition of a reference precipitation dataset that all alternative rainfall products (ERA5-P and LSTM outputs) are measured against, and try to replicate. In our previous review, we suggested the use of E-OBS as a training target, because this dataset is often used as a reference in the paper. In their response to our comment, the authors objected that "while E-OBS is superior to ERA5 in regions with dense data, using ERA5 has advantages in data-scarce regions". We agree with this statement, but believe that it relates to the applicability of the algorithm, which should not precede a thorough and logically robust testing in a controlled environment.

Overall, we repeat our request for the selection of a reference observed rainfall precipitation dataset to be used as a training target for the LSTM and computation of all performance metrics in the paper.

We have implemented the reviewer's suggestion by using the observational EOBS product as the target for our model runs. Since the original Caravan dataset (Kratzert et al., 2023) did not include the EOBS estimates for the training catchments; we preprocessed the data to derive the average precipitation estimate for each catchment. We then repeated our experiments using an ensemble network of three LSTM models (with different initialisation seeds) and report the mean results for both *with_discharge* and *without_discharge* runs. Our main finding remains largely the same in the new updated version.

In addition to looking at the gain over all the days, we also explored the performance gain over days with higher magnitude precipitation (shown in Figure 1). We could see that gains are considerably greater on days with higher recorded precipitation (increase in median NSE value of about 29% from 13% for days with more than 5 mm precipitation). This is logical because the discharge information is more effective in capturing extreme conditions. In contrast, the information gain is limited under average flow conditions.

[Figure]

**Figure 1 Comparison of performance gain for the with_discharge vs without_discharge models in NSE for different precipitation amounts. The first violin plot illustrates the average improvement across all days in the testing period. The second and third plots display the mean performance gains over the catchments, specifically focusing on days where precipitation exceeded 1 mm and 5 mm, respectively.**

For the continental analysis, we again calculated all the performance metrics and now compare both the *with_discharge* and *without_discharge* models to EOBS and ERA5 Land (Figure 2). The predictions from *without_discharge* model are also added for the out-of-sample analysis (Figure 3). For the out-of-sample predictions, we again observe that the LSTM estimate overestimates the EOBS value (new training target) in three out of four catchments; the runoff coefficients (Table 1) and timing of the peaks again point to the overall reliability of the estimate.

The forward hydrological model runs using HBV and CATFLOW were also repeated for the new estimate from the *with_discharge* model, and we again observed higher NSE values over the evaluation period compared to runs with ERA5 Land.

[Figure]

**Figure 2 The spatial patterns of the different time series metrics (Appendix C) mean wet day precipitation (MWD) – mm/day, 95th percentile limit (R95P) – mm/day, and Spearman autocorrelation values (SL) over the study catchments for the different precipitation estimates - ERA5 Land (top row): a) to c), *with_discharge* LSTM model (second row): (d) to (f), *without_discharge* LSTM model (third row): (g) to (i) and E-OBS (bottom row): (j) to (l) from 2006 to 2020 (2015 for CAMELS-GB catchments).**

[Figure]

**Figure 3 Precipitation estimates for flood events at four out-of-sample catchments: (a) Elsenz Schwarzbach, (b) Ernz, (c) Sueiro, and (d) Hoelzlebruck. The red line represents the observed daily streamflow, with a cross marking the day of the flood event. The orange curve indicates the precipitation predicted by the *with_discharge* LSTM model, while the green curve shows the precipitation predicted by the *without_discharge* model. The blue line reflects the original gauge-based EOBS time series, and the grey line represents the estimate from the ERA5 Land.**

**Table 1 Event characteristics (storm volume and runoff coefficients) for the four out of sample catchments**

| Event Characteristics | | Elsenz-Schwarbach | Ernz | Sueiro | Hoelzlebruck |
|---|---|---|---|---|---|
| **Precipitation (mm)** | ERA5 Land | 12.51 | 9.60 | 41.81 | 32.12 |
| | *with_discharge* | 32.79 | 42.75 | 58.53 | 50.85 |
| | *without_discharge* | 4.92 | 6.20 | 29.46 | 22.92 |
| | E-OBS | 20.07 | 51.72 | 29.50 | 44.90 |
| **Discharge (mm)** | | 5.98 | 26.88 | 23.39 | 19.14 |
| **Runoff Coefficient (-)** | ERA5 Land | 0.48 | 2.80 | 0.56 | 0.60 |
| | *with_discharge* | 0.18 | 0.63 | 0.40 | 0.38 |
| | *without_discharge* | 1.21 | 4.34 | 0.79 | 0.84 |
| | E-OBS | 0.30 | 0.52 | 0.79 | 0.43 |

Comment #2: The second fundamental comment made in our previous review related to the use of the mean catchment rainfall derived from ERA5-P as a predictor in the authors' statistical model. At the same time, ERA5-P is the training target of the author's statistical model. As a result, ERA5-P derived data are parts of both predictors and predictands. This is a fundamental flaw in statistical modelling, which cannot be accepted if aiming at publishing in a scientific journal such as HESS. At the same time, the authors have maintained the use of this predictor in their revised manuscript.

In their response, the authors first argued that this issue is relevant to real-time forecasting ("it is important to note that our end goal is different from an operational forecast"). We disagree with this view. The issue we are raising here relates to the problem of predicting output data while using part of this data as a predictor. This creates a risk of inflating model performance compared to a realistic use of the model, where the output data is not available, by definition (otherwise, a prediction model would not be required). In addition, this configuration prevents the model from being realistically used: mean catchment rainfall is one of the core data one would expect to extract from a rainfall product. If this data is required as part of the model inputs, we do not see much use of the LSTM rainfall product presented by the authors.

Overall, we repeat our request to remove mean catchment rainfall as a predictor in the paper. We believe this is extremely simple to do, as most computations have already been done by the authors.

We appreciate the reviewer's detailed comments regarding the possible short comings of using mean catchment rainfall derived from ERA5-P as a predictor. Our initial choice to include static attributes was based on incorporating climatic indicators relevant to our catchments, and it made sense to include mean precipitation. Additionally, we were influenced by other hydrological modelling studies (Gharari et al., 2021) that utilised average runoff information to enhance calibration for the traditional streamflow prediction problem.

After considering the detailed concerns raised by the reviewer and the editor regarding the statistical modelling setup that used LSTM, we have decided to exclude *pmean* from the paper. We have re-run all our model simulations with the new setup, which does not include *pmean* as a predictor. The results align with previous studies (Heudorfer et al., 2024; Li et al., 2022) that suggest the physiographic characteristics of static features may be irrelevant; what truly matters is the presence of unique identifying information.

**References**

Gharari, S., Gupta, H. V., Clark, M. P., Hrachowitz, M., Fenicia, F., Matgen, P., & Savenije, H. H. G. (2021). Understanding the Information Content in the Hierarchy of Model Development Decisions: Learning From Data. *Water Resources Research*, *57*(6). https://doi.org/10.1029/2020WR027948

Heudorfer, B., Gupta, H., & Loritz, R. (2024). *Deep Learning Models in Hydrology Have Not Yet Achieved Entity Awareness*. https://doi.org/10.22541/essoar.172927199.90156076/v1

Kratzert, F., Nearing, G., Addor, N., Erickson, T., Gauch, M., Gilon, O., Gudmundsson, L., Hassidim, A., Klotz, D., Nevo, S., Shalev, G., & Matias, Y. (2023). Caravan - A global community dataset for large-sample hydrology. *Scientific Data*, *10*(1), 61. https://doi.org/10.1038/s41597-023-01975-w

Li, X., Khandelwal, A., Jia, X., Cutler, K., Ghosh, R., Renganathan, A., Xu, S., Tayal, K., Nieber, J., Duffy, C., Steinbach, M., & Kumar, V. (2022). Regionalization in a Global Hydrologic Deep Learning Model: From Physical Descriptors to Random Vectors. *Water Resources Research*, *58*(8). https://doi.org/10.1029/2021WR031794

---

## Author Response (AR3)

**Can discharge be used to inversely correct precipitation? (hess-2024-375)**

Ashish Manoj J, Ralf Loritz, Hoshin Gupta, Erwin Zehe

Dear Dr. Roger Moussa,                                                        20.10.2025

Attached, please find the revised version of the manuscript "*Can discharge be used to inversely correct precipitation?*" co-authored with R. Loritz, H. Gupta and E. Zehe, to be considered for publication in Hydrology and Earth System Science.

After carefully reviewing the minor comments from Anonymous Reviewer 1 in the previous round, we have decided to implement the changes suggested by the reviewer in the revised manuscript. We have omitted the ambiguous usage of the term *effective precipitation* and changed it to *actual precipitation*.

We would like to thank the Editor and Anonymous Reviewer 1 again for giving us another opportunity to revise our manuscript.

Please get in touch with me if you need any additional information.

Thank you very much for your consideration.

Best regards,

Ashish

On behalf of Ralf, Hoshin and Erwin

Email: ashish.jaseetha@kit.edu

**Reviewer 1:**

The authors would like to thank Anonymous Reviewer 1 for again carefully reviewing our manuscript and providing their helpful comments. We have followed the reviewer's suggestion in the revised manuscript. The following responses have been prepared to address all the reviewers' comments point-by-point. We have responded (in black) to the reviewer's comment (in blue).

**General comments:**

Dear authors,

Thank you for improving the manuscript. I basically just have one topic left to be clarified, which is the use of the term "effective precipitation".

Line 166:

"Both models were trained to predict daily catchment average precipitation sums from the observational EOBS product (ERA5 Land). Therefore, we only deal with spatially averaged timeseries for precipitation, assuming that these values represent the effective precipitation over the entire catchment."

In my first review I already pointed out the ambiguous use of the term "effective precipitation". I post here my comment and your response:

I agree that we can never really know the "true" precipitation amount a catchment receives. However, the term effective precipitation is a major term in hydrology and I have the feeling you are using it unconventionally: it describes the fraction of precipitation that is converted into run off, after a fraction is lost to interception, infiltration etc.. Except for completely sealed surfaces the effective precipitation is always smaller than the precipitation. Are you mixing up the terms of, what you describe as "true precipitation" and effective precipitation?

Keeping this in mind, I find the sentence in line 166 quite confusing. EOBS and ERA5 land contain precipitation and the LSTMs are trying to predict this precipitation, not the effective precipitation.

We thank the Reviewer for pointing out the ambiguity in our usage. Line 166 has been updated to *'assuming that these values represent the  actual precipitation over the entire catchment'.* The term has also been omitted from Line 353 in the revised manuscript.